# Underlying drivers of coral reef vulnerability to bleaching in the Mesoamerican Reef
Aarón Israel Muñiz-Castillo [1,2] ✉, Andrea Rivera-Sosa [1], Melanie McField [3,4] ✉, Iliana Chollett[5], C. Mark Eakin[6], Susana Enríquez [7], Ana Giró [8], Ian Drysdale[9], Marisol Rueda[2], Mélina Soto [2], Nicole Craig[10] & Jesús Ernesto Arias-González [1] ✉

Coral bleaching, a consequence of stressed symbiotic relationships between corals and algae, has escalated due to intensified heat stress events driven by climate change. Despite global efforts, current early warning systems lack local precision. Our study, spanning 2015–2017 in the Mesoamerican Reef, revealed prevalent intermediate bleaching, peaking in 2017. By scrutinizing 23 stress exposure and sensitivity metrics, we accurately predicted 75% of bleaching severity variation. Notably, distinct thermal patterns—particularly the climatological seasonal warming rate and various heat stress metrics—emerged as better predictors compared to conventional indices (such as Degree Heating Weeks). Surprisingly, deeper reefs with diverse coral communities showed heightened vulnerability. This study presents a framework for coral reef bleaching vulnerability assessment, leveraging accessible data (including historical and real-time sea surface temperature, habitat variables, and species composition). Its operational potential lies in seamless integration with existing monitoring systems, offering crucial insights for conservation and management.

The Anthropocene era is underway and is characterized by rising ocean temperatures and increased exposure to heat stress events resulting in widespread coral bleaching[1,2]. One of the largest bleaching events to affect coral reefs worldwide occurred during 2014–2017[1–5]. During and after this long-lasting bleaching event, multiple ecological consequences were observed, such as loss of coral cover and diversity, changes in species composition, and reductions in coral growth and recruitment[2,6–8]. The increasing frequency and intensity of mass coral bleaching events call for an improved system to evaluate the vulnerability of reefs to heat stress, based on the biological species composition and exposure to different measures of heat stress and thermal variation. Only by improving the evaluation of the main drivers and metrics behind coral bleaching will it be possible to develop effective conservation strategies for this valuable ecosystem[9–11]. Therefore, an accurate prediction of the vulnerability of bleaching is a critical aspect of the management and applied conservation of coral reefs.

The application of a bleaching vulnerability framework allows the identification of exposure to the level of stress (frequency and magnitude of heat stress) and reef sensitivity metrics (intrinsic ecological characteristics that affect the potential impact of heat stress on a particular reef) needed to predict the severity of coral bleaching better[12]. Heat stress indicates the exposure component of reef vulnerability to coral bleaching. The most widely used descriptor is the NOAA Degree Heating Weeks (DHW) product, although other metrics to measure heat stress accumulation have been proposed[1,3,4,13–16]. The NOAA and other operational warning products lack an indicator of reef sensitivity to heat stress, several of which have been proposed, such as recent thermal trajectories[17–19] and thermal variability[3,9,15,16,20,21]. Other intrinsic components of the coral response to thermal stress may also provide insights into the sensitivity/resistance of the coral community to heat stress, which allows us to generate sensitivity indices for different coral species[3,16,22,23]. Additionally, the response of individual coral colonies to heat stress can vary due to factors such as the

[1]Laboratorio de Ecología de Ecosistemas de Arrecifes Coralinos, Departamento de Recursos del Mar, Centro de Investigación y de Estudios Avanzados del Instituto Politécnico Nacional. Mérida, 97310 Yucatán, Mexico. [2]Healthy Reefs for Healthy People, Puerto Morelos, Mexico. [3]Healthy Reefs for Healthy People, Fort Lauderdale, USA. [4]Smithsonian Marine Station, Smithsonian Institution, Fort Pierce, 34949 FL, USA. [5]Sea Cottage, Louisburgh, Co. Mayo, Ireland. [6]Corals and Climate, Silver Spring, 20904 MD, USA. [7]Laboratorio de Fotobiología. Unidad Académica de Sistemas Arrecifales Puerto Morelos, Instituto de Ciencias del Mar y Limnología, Universidad Nacional Autónoma de México, Cancun, 77500 Quintana Roo, Mexico. [8]Healthy Reefs for Healthy People, Guatemala City, Guatemala. [9]Healthy Reefs for Healthy People, Tegucigalpa, Honduras. [10]Healthy Reefs for Healthy People, Belmopan, Belize. ✉e-mail: israel@healthyreefs.org; mcfield@healthyreefs.org; earias@cinvestav.mx

symbiotic algal community[24,25], coral optical properties[26–28], the potential contribution of the coral phenotype[29], and the colony or skeleton morphology[28,30]. On the other hand, the physical and oceanographic characteristics of reefs are also important determinants of coral bleaching variation, including depth[16,31–33], local influence of turbidity, exposure to currents, or nutrient enrichments from upwelling[34]. Even the geographic position or region in which reefs are located may be a good predictor of coral bleaching, highlighting the potential spatial variation in stress exposure or adaptation processes in corals across a geographical gradient[3,15,19,21,35]. Integrating all this information may enhance our ability to predict bleaching, supporting actions that coral reef managers and stakeholders need to take to protect coral reef ecosystems[3,14,16,36–38]. Significant advances have been made in predicting the spatial variation of coral bleaching at regional and global scales[3,14–16,21,35,39]. However, most existing efforts focus on predicting and recognizing the drivers of bleaching during a single event or in different isolated events[3,15,16,21,35]. The increasing role of consecutive heat stress events is of vital importance, as we have recently experienced recurring extreme heat events[2,18,19]. In the Caribbean, there are no predictive frameworks designed to evaluate the cumulative effects of consecutive heat stress events or to allow the additive inclusion of other stressors that can help explain the spatial variability of coral bleaching events.

In this study, we present an advanced multidimensional framework for assessing coral reef vulnerability to bleaching. Given the backdrop of prolonged, high-intensity heat stress events over several years, we posit that emerging thermal patterns and intrinsic reef characteristics may significantly influence the observed variations in coral bleaching. To evaluate this hypothesis, we analyzed remote sensing data and 266 in situ reef level samples/observations recorded during the seasonal 'bleaching window', August to December (Supplementary Fig. 1) along the Mesoamerican Reef (MAR)[40], during 2015–2017. During this period, MAR reefs were exposed to high levels of heat stress[40]. Our emphasis is on unraveling the relationships among 23 different metrics of stress exposure (heat stress exposure metrics) and sensitivity (intrinsic climatological thermal patterns, coral species composition, coral species diversity, and depth) to coral bleaching. In the analysis, we describe first the temporal and spatial patterns of bleaching severity. Then, we analyze the association of these patterns with the 23 exposure and sensitivity metrics selected to test their capacity to predict coral bleaching severity. Gradient boosted models (GBM, also known as Boosted Regression Trees), a machine learning algorithm, were used to identify relative direct associations, non-linear relationships, and interactions[3,41–43]. Our results uncovered the additive effect of seven metrics to explain ~75% of the variation of coral bleaching severity. Based on these results, we propose a model to predict coral reef vulnerability to bleaching, with a high potential for use as an early warning system for the Mesoamerican Reef transferable to other ecoregions in the wider Caribbean. It can also be used to predict reefs with intrinsically more resistance to bleaching for conservation planning purposes. The operational use of our model is facilitated by the accessibility of essential data sources, including historical and actual sea surface temperature data, habitat variables, and species composition information. These readily available datasets enable efficient implementation and integration into existing monitoring systems, providing valuable insights for coral reef conservation and management efforts.

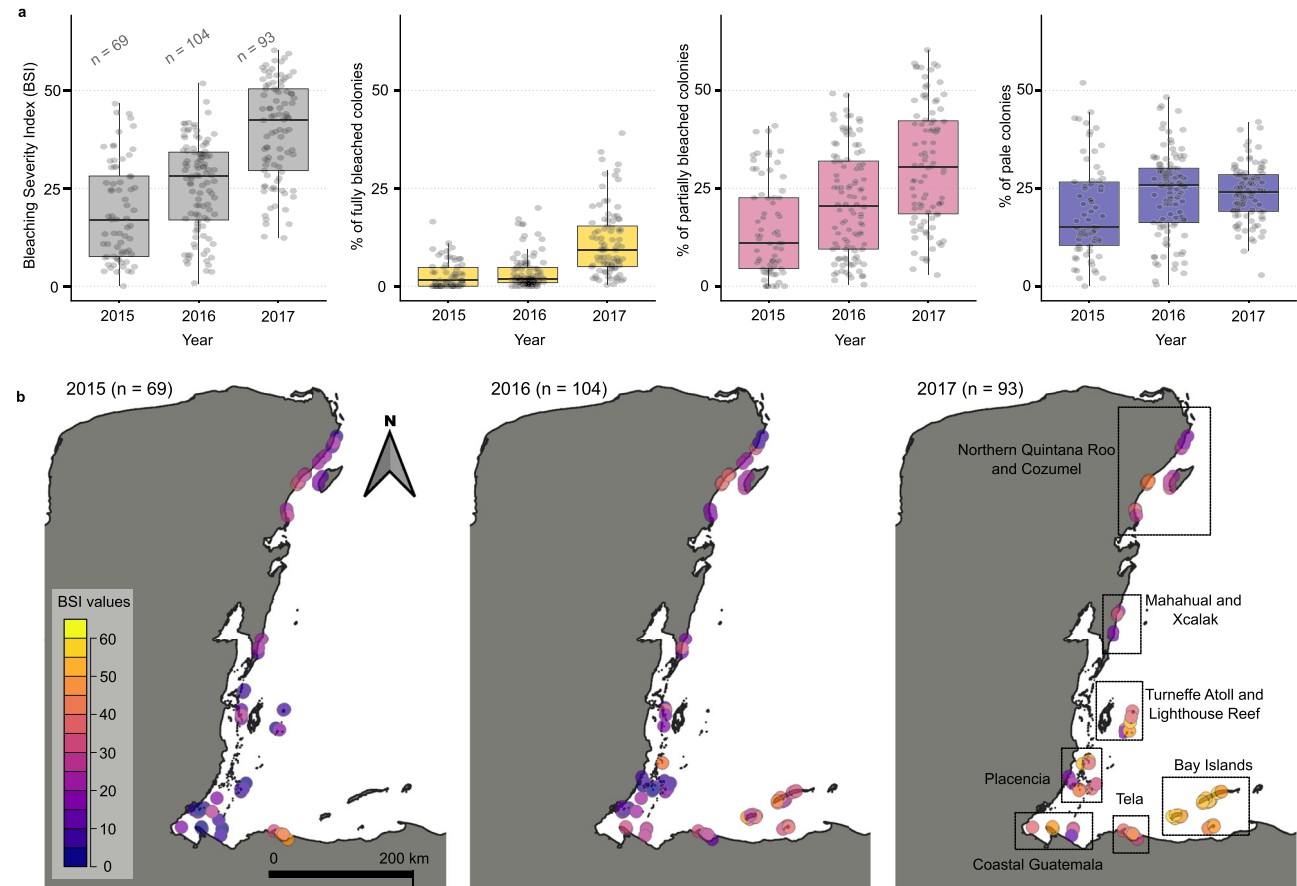

**Fig. 1 | Spatiotemporal variation of coral bleaching in the Mesoamerican Region during the years 2015–2017. a** Distribution of the bleaching severity index (BSI) values and the proportion of colonies in each bleaching category, each year, for all the reefs sampled. Information on the BSI's statistical descriptors and the categories defined for coral bleaching can be found in Supplementary Table 1. **b** Maps illustrate the spatial distribution of the severity of coral bleaching in each year, for all sampled reefs.

## Results and discussion

### Bleaching patterns in the MAR reefs during 2015–2017

Coral reefs along the MAR paled or bleached moderately during 2015–2016 and bleached substantially in 2017 (Fig. 1). The bleaching severity index used here incorporates coral response categories that have previously been used in the MAR and globally[3,22,44,45]. The bleaching severity is calculated from the sum of the proportion of colonies in each response category, weighting each category according to its ecological impact, with the proportion of fully bleached colonies having a higher weight and the proportion of pale colonies having a lower weight. We assessed 55,177 colonies using the following categories: pale (with significant discoloration), partially bleached (bleached tissue present), and fully bleached (with > 90% bleached tissue). Over half of the 266 reef-level samples/observations recorded from 2015 to 2017 reached bleaching severity values ≥ 28.0 (i.e., reefs with at least 28% affected colonies; Fig. 1a). The most common categories were partially bleached and pale (Fig. 1a; Supplementary Table 1). During 2015–2016, we observed moderate bleaching (Fig. 1a, b), localized in certain regions such as Tela Bay, the Bay Islands of Honduras, and some reefs in northern Quintana Roo, Mexico (Fig. 1b). However, in 2017 bleaching was more severe (Fig. 1a; Yuen's test for dependent samples, $p < 0.001$; Supplementary Table 2) and widespread, affecting a large MAR reef area (Fig. 1b). The observed patterns were consistent with an increase in heat stress exposure over 2015–2017. About 50% of the MAR reefs were exposed to severe heat stress in 2017, with DHW values higher than 7 °C-weeks[40]. Moreover, the severity of bleaching recorded in 2017 revealed an extensive coral bleaching event. Previous events documented for the MAR occurred in 1995[45], 1998[46], 2005[13], and 2010[47]. The 2017 heat stress event was the most intense in the region until that date[40], with more than 25% of the reefs presenting over 15% of fully bleached coral colonies. This result contrasts with previous observations, whereas 6% of sites presented whole bleaching in Belize during the 1995 event[45]. This impact may have been similar to past bleaching events, such as 2005, with reports of 28% of coral colonies affected in Mexico and Belize[13].

### Drivers of coral bleaching in the MAR reefs

We found that key heat stress exposure metrics in combination with intrinsic factors such as climatological thermal variability, and reef sensitivity based on species composition, depth, and coral diversity are strong predictors of coral bleaching severity (Table 1; Fig. 2). Although we recognize our limitation in not including other variables (such as sedimentation, nutrient enrichment, pollution, and changes in water circulation patterns) known to also contribute the impact of heat stress on symbiotic corals[3,15,21,31,34,39,48–51]. GBM (Supplementary Fig. 2) showed that a combination of seven noncollinear metrics of stress exposure and reef sensitivity explained 75% of the spatial variation of bleaching severity (Fig. 2a). The models obtained did not show signs of spatial autocorrelation (Fig. 2b) or atypical residuals patterns (Fig. 2c). In the model, the strength of interactions was relatively low (H-statistic < 0.25), which represents a component below 25% of the standard deviation due to the interaction between tested pairs of predictors (Supplementary Table 3). This finding supports a dominant additive effect among the identified metrics because the interactions did not represent a major source of variation in the model. Our results showed that the climatic thermal variation represented by the ROTC$_{clim}$ indicator (termed rate of seasonal warming in Chollet et al.[9]) is a thermal metric that considerably influences the severity of bleaching. ROTC$_{clim}$ combined with the heat accumulated in the last 28 days (HS$_{28days}$) before bleaching observation, accounts for about 50% of the relative contribution in our model. Different novel thermal and cumulative heat stress patterns (in different time windows and with different stress thresholds) are better predictors of actual coral bleaching and could provide improved predictive capacity to managers in the MAR and globally in the current situation.

We found the climatological seasonal-warming rate (sensu Chollet et al.[9]) from 1985 to 2012 (ROTC$_{clim}$) to be the main driver of the severity of coral bleaching (Fig. 2a). ROTC$_{clim}$ expresses the mean trend of temperature change over the spring-to-summer transition (i.e., during the three months before the maximum weekly average temperature) from 1985 to 2012. The

reefs with the highest ROTC$_{clim}$ had the lowest bleaching severity (Fig. 2d). Our findings suggest that corals in reefs with a thermal history characterized by higher and more rapid seasonal change were more resistant during the bleaching event. Environmental seasonality is particularly important for corals, as seasonal variation in irradiance and temperature directly impacts the rate of photosynthetic carbon fixation. In contrast, temperature changes also induce important adjustments in the heterotrophic metabolism. Corals acclimated to areas with moderate to high thermal variations are accustomed to physiological variations and adaptations which may assist their ability to acclimate to thermal stress-associated bleaching events[15,16,21,52–54]. Several Caribbean corals have demonstrated an ability to express two different coral holobiont phenotypes, winter, and summer, with contrasting susceptibility to bleaching under similar heat stress exposure[29]. The rate of temperature change in spring to early summer may determine when the summer coral phenotype is fully achieved and, therefore, when the coral is prepared to cope with heat stress[29]. This implies that a more robust response could be expected when accumulated heat stress coincides with the complete expression of the summer phenotype[29]. However, when the heat stress event occurs before the complete expression of the summer coral phenotype, or if moderate or acute heat stress is prolonged for too long, different physiological responses can be expected[29]. This could be a physiological mechanism that could explain the results of our work. Still, more research is needed to explain the contribution of ROTC$_{clim}$ to coral bleaching during heat stress events. This is of relevance as it could represent an emerging pattern of potential resilient populations, which may modify the perception we have regarding previous existing models[9,16,55], especially if this is an emerging pattern in all reefs worldwide. In our results, the ROTC$_{clim}$ represents a climatological thermal variability metric without collinearity with most of the indicators of heat stress. It is important to mention that high rates of temperature change do not necessarily result in high exposure to heat stress, and sites with high rates of temperature change but low heat stress could be considered as potential refugia[9,15,52,56].

Our results also revealed that the second most important predictor was the 28 accumulated days of heat stress (HS$_{28days}$; Fig. 2a). Bleaching severity presented a positive non-linear association with HS$_{28days}$, and reefs exposed to more than 15 °C of accumulated heat in the last 28 days were the most affected (Fig. 2d). It is worth mentioning that HS$_{28days}$ was a better predictor of bleaching severity than the most common indicator of heat accumulation over three months, the degree heating weeks (DHW)[37]. DHW quantifies heat stress accumulation by summing positive anomalies above 1 °C for 84 days and dividing by 7 to express values per week[37]. However, previous studies have demonstrated that the prediction of coral bleaching can be improved for some locations by testing different temporal windows and thresholds of accumulated heat stress[14,16,23,33,39,57]. Indeed, some of these studies have found that the use of heat stress metrics that consider time windows of one or two months can significantly improve the prediction of coral bleaching in the wider Caribbean reefs[23,33,39]. Coral bleaching is a dysfunctional physiological response that requires a certain exposure to heat and light that can be attained in different periods[27,29,58–62]. Thus, it is important to consider different potential temporal windows can result in the same level of heat stress accumulation. Experimental studies have shown that coral bleaching can be seen immediately after acute heat stress or may require one or two months of accumulated heat stress to manifest[29,58–62]. However, coral recovery after bleaching can be observed in periods of 20 days to 2 months[45,58,60–63]. Additionally, the increasingly frequent marine heat waves and changes in seasonal warming patterns in the Caribbean are potentially promoting accumulated heat stress over shorter periods than in previous decades[64], which highlights the importance of these types of metrics and the use of different time windows for estimating accumulated heat[57]. Complementary metrics to measure heat stress accumulation can be useful for the continued improvement of early warning systems, and to advise emergency response strategies[57].

In addition to HS$_{28days}$, we also identified two other metrics derived from DHW as important predictors of bleaching (Fig. 2a). Sites with a long-term increase in DHW (from 1985 to the sampled year) were found to have

**Table 1 | Seventeen remote sensing and 6 field-based variables considered as potential predictive drivers of coral bleaching divided by (a) selected metrics and (b) metrics not selected in the final gradient-boosted model**

| Variable | Description | Range of values (units) | Notes |
|---|---|---|---|
| *(a) Metrics selected in the final gradient-boosted model* | | | |
| Depth | The mean depth of the reef in meters. | 1.00–26.45 (m) | The sixth most relevant variable in the model. BSI was higher at greater depths. |
| Diversity | The diversity of corals calculated from Hill's number one is equal to Shannon's diversity exponent, this represents true diversity without considering the less abundant or "rare" species[84]. | 2.173–21.212 (effective species) | The seventh most relevant variable in the model. BSI was higher at greater diversity. |
| $HS_{28days}$ | The sum of HS in 28 days before the sampling date. | 0.0–48.55 (°C) | The second most relevant variable in the model. BSI was higher in reefs with higher heat stress in the last 28 days. |
| $SI_{reef}$ | Reef sensitivity is based on the relative abundance of species weighted by species response to bleaching[3,23]. | 0.1438–0.3665 (index value) | The third most relevant variable in the model. A sharp increase in the BSI as the sensitivity index increases. |
| $ROTC_{clim}$ | The climatological seasonal-warming rate ($ROTC_{clim}$) is the average of the annual ROTC values for the period 1985–2012 (*sensu* Chollett et al. 2014)[9]. The seasonal warming rate (ROTC) reflects the trend in temperature change over 84 weeks during summer[9]. | 0.1931–0.2464 (°C by week) | The most relevant variable in the model. Negative relationship with BSI. Reefs with a higher rate of climatological seasonal warming rate exhibited less bleaching. |
| Trend of DHW | DHW quantifies heat stress by summing up positive daily anomalies above 1 °C above the MMM over 84 days (12 weeks), divided by 7 to express values per week[38]. 'Trend of DHW' is the trend of annual maximums DHWs from 1985 to the sampling year, a trend obtained from a generalized least square model[42]. | 0.00000–0.30064 (°C-weeks by year) | The fifth most relevant variable was selected in the model. Higher BSI in reefs with a higher rate of increase in DHWs. |
| Δ DHW | Difference between the maximum observed value of DHW in the current event up to the sampling date and the maximum observed value of DHW in the last year (building on Hughes 2019[19]) | −3.7787 to 7.7007 (°C-weeks) | The fourth most relevant variable was selected in the model. Higher BSI in reefs with higher recent heat stress (ΔDHW ≥ 6). |
| *(b) Metrics not selected in the final gradient-boosted model* | | | |
| DCA1 | The first axis in a multidimensional ordination analysis based on species composition and applying a Detrended Correspondence Analysis[85]. Considered a potential ecological gradient related to bleaching[3]. | −1.6264 to 2.2710 (axis position) | Not selected in the final model but high correlation with diversity. |
| $DHW_{28days}$ | DHW calculation considering a 28-day window[16]. | 0.0000−6.9358 (°C-weeks) | Not selected in the final model but high collinearity with different heat stress metrics. |
| DHW | Conventional DHW calculation considering an 84-day window[38]. | 0.5141–13.7115 (°C-weeks) | Not selected in the final model but high collinearity with different heat stress metrics. |
| $DHW_{555days}$ | DHW calculation considering a 555-day window represents the accumulated stress since the beginning of the summer of the previous year. | 1.217–32.874 (°C-weeks) | Not selected in the final model but high collinearity with different heat stress metrics. |
| $HS_{3days}$ | Hotspots (HS) represent positive daily anomalies above the MMM[38]. $HS_{3days}$ is the average HS in the last three days before the sampling date. | 0.00000–1.49589 (°C) | Not selected in the final model but high collinearity with different heat stress metrics. |
| $HS1_{28days}$ | Number of days with HS values greater than 1 °C in the last 28 days[16]. | 0–28 (days) | Not selected in the final model but high collinearity with different heat stress metrics. |
| $HS1_{84days}$ | Number of days with HS values greater than 1 °C in the last 84 days[16]. | 3–60 (days) | Not selected in the final model but high collinearity with different heat stress metrics. |
| $HS2_{28days}$ | Number of days with HS values greater than 2 °C in the last 28 days[16]. | 0–4 (days) | Not selected in the final model but high collinearity with different heat stress metrics. |
| $HS2_{84days}$ | Number of days with HS values greater than 2 °C in the last 84 days[16] | 0–4 (days) | Not selected in the final model but high collinearity with different heat stress metrics. |
| $HS_{84days}$ | The sum of HS in 84 days before the sampling date. | 19.94–108.30 (°C) | Not selected in the final model but high collinearity with different heat stress metrics. |
| $HS_{555days}$ | The sum of HS in 555 days before the sampling date. | 90.13–332.12 (°C) | Not selected in the final model but high collinearity with different heat stress metrics. |
| MMM | Monthly average of the hottest month registered during 1985-2012[38] | 28.62–29.06 (°C) | Not selected in the final model but high correlation with some thermal metrics. |
| RFI | Reef Functional Index based on the relative abundance of species multiplied by a functional coefficient[71] | 0.1800–0.7679 (index value) | Not selected in the final model. |
| Richness | Number of coral species. | 5–34 (species) | Not selected in the final model but high collinearity with diversity. |
| $ROTC_{84days}$ | Maximum ROTC observed 84 days before sampling date[9]. | 0.1667–0.3138 (°C by week) | Not selected in the final model but high collinearity with $ROTC_{clim}$. |
| $SD_{84days}$ | Standard deviations of the SST in the previous 84 days. | 0.2091–1.1063 (°C) | Not selected in the final model. |

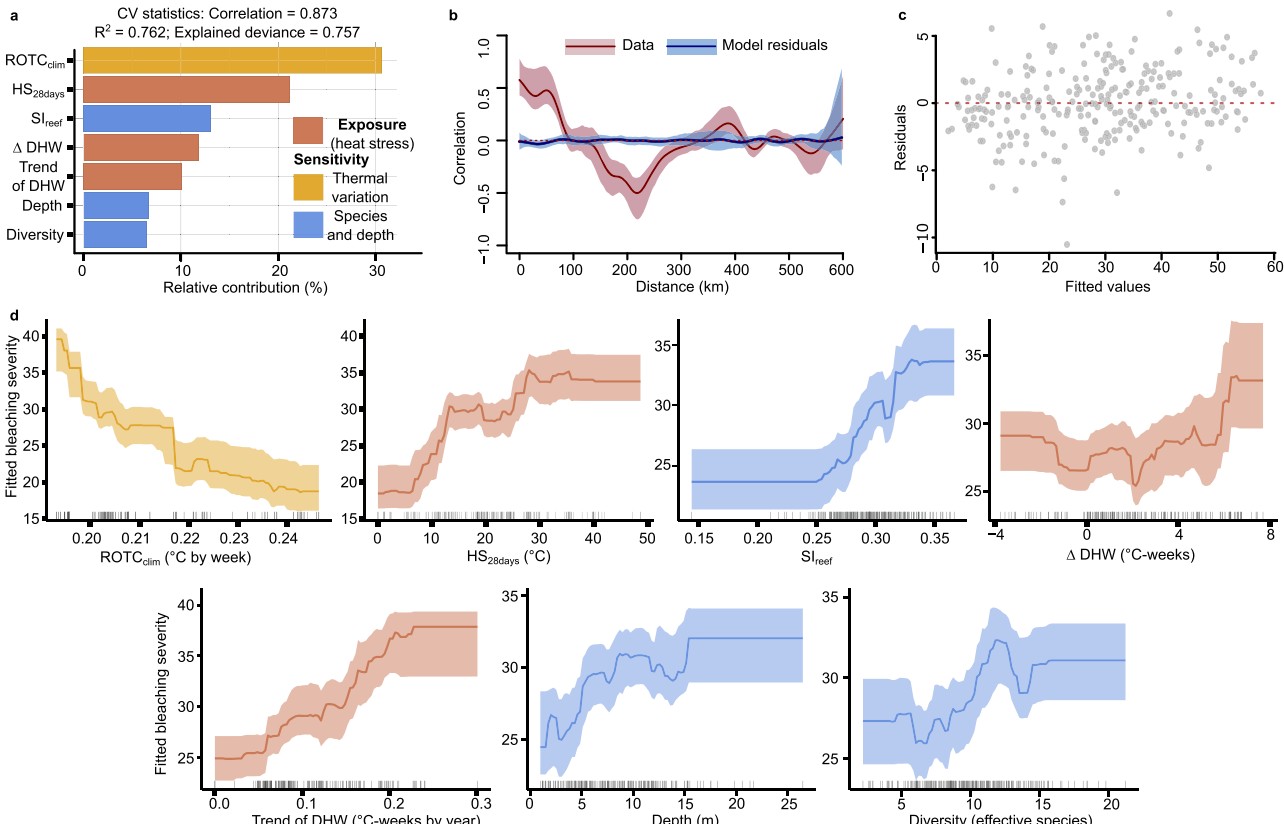

**Fig. 2 | Main drivers of coral bleaching and the identified associations of variation. a** Relative contribution of the key drivers identified using gradient boosted models (GBMs). **b** Spline correlogram displaying the spatial autocorrelation of the bleaching severity index values (data) and the model residuals. The *x*-axis represents the distance between sampling points in kilometers, regardless of direction. **c** Plot of residuals vs fitted values. **d** Plots of partial dependence of variables were selected in each of the GBMs. The shadow-colored area on the plots represents the 95% confidence interval generated from a bootstrap approach with 1000 permutations. Black hash marks on the x-axes represent the values of the surveyed reef data. ROTC_{clim} expresses the mean trend of temperature change during the spring-summer transition (i.e., 3 months before the maximum weekly average temperature) along the period 1985–2012, also termed rate of seasonal warming[9]; HS_{28days} is the sum of

positive anomalies in 28 days before the sampling date; SI_{reef} is the reef sensitivity based on the relative abundance of species, weighted by species response to bleaching; ΔDHW is the difference between the maximum observed value of DHW in the last year and the maximum observed value of DHW in the current event up to the sampling date; 'Trend of DHW' is the trend of annual maxima DHWs since 1985 to the sampling year; Depth is the mean depth of each reef; diversity is the Hill's number one expressed in effective species. Under this framework of Hill's numbers, the diversity of a community is measured as the effective number of species in it, which can be understood as the number of species in a virtual, perfectly balanced community, in which all species are equally common, and in which the average relative abundance of the species in the real community is conserved[80].

greater severity of bleaching (Fig. 2d). Plotting this DHW annual trend[40] can reveal reef areas where the heat stress events have increased in severity in recent years (i.e., higher trends in DHW). Our results are consistent with previous studies showing that long-term increases in SST or frequency of heat stress anomalies are associated with increases in coral bleaching[1,21]. We also observed that the recent heat stress history, represented by the difference between the most recent and previous year in the maximum DHW values (ΔDHW)[18], was an important metric for predicting bleaching severity. The highest severity of bleaching was observed in sites with ΔDHW values between 4 to 8 °C-weeks (Fig. 2d). High values of ΔDHW indicate much greater heat exposure during the recent event compared to the previous year, which corresponded to a considerable increase in bleaching severity, particularly at values above 6 °C-weeks. This aligns with the findings of Hughes et al.[18], who demonstrated that the severity of bleaching in 2017 on the Great Barrier Reef was significantly influenced by the geographic patterns of bleaching observed in 2016. The study showed that reefs with high heat exposure in 2016 exhibited reduced bleaching in 2017, even under similar thermal stress, suggesting an ecological memory effect. This response may result from acclimatization, adaptation, or changes in species composition, highlighting the potential physiological or ecological memory retained by corals due to past heat stress[18,19]. For example, after a site's first significant bleaching event, the most sensitive species would be reduced in

relative abundance, leaving a higher abundance of species that can better resist the next heat stress event[18,65]. In addition, at the level of an individual coral, it has also been documented that high previous exposure to heat stress can make some coral species more resistant to the next heat stress event[5,61]. However, the ability of a coral to build resistance will depend on the metabolic costs incurred during the stress event and on its ability to fully recover before the next heat stress event occurs. Thus, for a certain range of stress, some corals can have a negative response to consecutive events, and they will not achieve acclimatization[27,29,61,63]. Considering the existing evidence and the results obtained in this analysis, we conclude that not only the current thermal regime but also the history of heat stress of a particular reef needs to be considered to predict the risk of bleaching during a new event. In summary, the long-term, and recent history of exposure to heat stress is fundamental for the prediction of coral bleaching during long-lasting events.

Among the variables not associated with temperature; the best predictor of bleaching severity was the index of reef sensitivity (SI_{reef}; Fig. 2a), which is based on the species-specific response to bleaching (Fig. 3). SI_{reef} is calculated as the sum of the relative abundance of each coral species multiplied by its species-specific bleaching impact observed during the 2015–2017 events (Fig. 3). While this index was derived from data on bleaching responses during these years for the whole MAR, it was also used

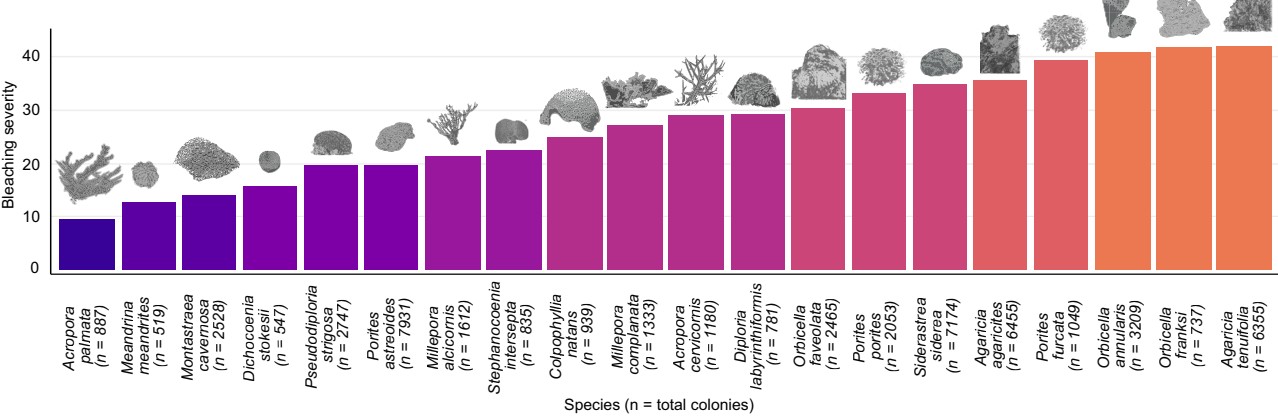

**Fig. 3 | Bleaching Severity Index in the most abundant coral species.** The figure shows only the species with more than 500 colonies assessed during the entire period (2015–2017). The information on the overall impact observed on each species is shown in Supplementary Table 4.

to assess how the composition of species with different sensitivities could predict overall reef bleaching severity in the same period, acknowledging the circular nature of using observed data to define and then test sensitivity. The standardized indicators of species sensitivity to bleaching, such as metrics based on the overall response of corals to different bleaching events, can be useful parameters for the development of conservation strategies on coral reefs and for predicting the severity of coral bleaching on a particular reef[3,22,23,66]. We found a positive association between $SI_{reef}$ and bleaching severity (Fig. 2d). The most affected reefs were those dominated by sensitive species, such as those of the genera *Agaricia* and *Orbicella*, or species such as *Siderastrea siderea* and *Porites furcata* (Supplementary Fig. 3). All these species showed a high degree of bleaching severity (≥30; Fig. 3). Our species-specific quantification of bleaching severity matches the patterns already documented for the wider Caribbean reefs[23,33,45,67]. However, it does not agree with some previous findings[28,30]. Species with thin tissues and plated colony morphologies such as those of the genus *Agaricia*, were, as expected, among the most sensitive, while the branching *Acropora palmata* showed the lowest bleaching severity within the most abundant species (Fig. 3). It is important to highlight that some species, especially in the genus *Acropora*, have been exposed to constant selective pressures and a decrease in populations during the last decades, therefore the remaining colonies may be more resistant, as the bleaching observed in these taxa was minor[68,69]. Earlier studies of symbiont clade bleaching sensitivity suggested *Acropora* species with clade A would be less sensitive[70], and this was supported by field evidence in the first 1995 bleaching event in Belize[45]. Meandroid morphs and species of the families Siderastreae and Poritidae, which may be considered resistant to coral bleaching[28,30], were among the most sensitive in this field study observing elevated values of BSI in these species (Fig. 3). Previous studies concluded that coral morphologies more efficient at collecting light (such as branched *Acropora*) may be more sensitive to light stress[28]. Attempts to parameterize this species variability with indicators such as the reef functional index (RFI)[71], showed low informative value in our study. Our findings also highlighted the need for a better understanding of the cellular and functional mechanisms behind the physiological disturbance of this symbiotic association and all the biological processes that can confer stress resistance at the symbiont, host, and holobiont/microbiome levels. Multiple factors influence these processes, highlighting the importance of basing analyses on observations and not generalizing theoretical or experimental results applied to different scales and regions. The exact combination of stressors may vary with each bleaching event in space and time.

Finally, the two last predictors identified as relevant were depth and coral species diversity (Hill's N1 equal to Shannon's diversity exponent; Fig. 2a). The model showed that the severity of bleaching on reefs was higher with increasing depth, whereas reefs with lesser bleaching severity occurred at depths shallower than five meters (Fig. 2d). Thus, our results contrast with the "depth refugia" hypothesis[31,32], which postulates that deep reef areas will be less affected by coral bleaching due to heat and light attenuation, and agrees with other studies that have found no clear depth refuge for coral species[32,33,72–74]. Corals in deep areas are less prone to acclimatory environmental pressure in contrast to corals in shallower reefs, causing deeper corals to be more affected by heat stress and high-temperature variation when exposed, and may have had fewer opportunities in the past to develop resistance to heat stress[15,33,53,73]. Increases in bleaching severity were also observed in the most diverse reefs, while lower values were detected in biodiversity levels of five to ten effective species (Fig. 2d). This pattern could be explained by the impact of past disturbances on coral diversity, as more diverse reefs may have experienced the lowest impact of past disturbances[75–77], and therefore be more affected during the severe events of 2014–2017. Exposure to heat stress has been associated with a loss of coral diversity in some reefs in the wider Caribbean[78]. Unfortunately, few studies have linked reef diversity with reef sensitivity to bleaching. Our results support previous data that have documented a positive association between coral diversity and bleaching severity at the reef or regional scales[79], and conflicts with studies that have suggested there is a negative association between diversity and bleaching severity at a global scale[21]. This suggests that coral diversity does not necessarily confer protection at the regional scale and is more likely to be an attribute associated with the prior disturbance history. We observed that the severity of bleaching was high in reefs with low diversity but dominated by sensitive species such as *Agaricia tenuifolia* and *Porites porites*, whereas low diversity reefs dominated by more resistant species such as *Porites astreoides* were among the least affected (Supplementary Fig. 3). Also, more diverse reefs could have more abundance of different sensitive species. Our results then stress the relevance of multiple metrics and the integration of different approaches (i.e., remote sensing, coral physiology, and ecological surveys) to fully understand the range of coral responses to heat stress and the risk of bleaching on a particular reef. The understanding of why shallower reefs with lower diversity could be less vulnerable to bleaching, or how they handle high environmental variability or form more resistant coral communities after surviving greater past disturbances (ecological filters), requires a complex and integrative approach.

## Conclusions

In conclusion, our study revealed key drivers of coral bleaching by integrating various reef sensitivity and heat stress metrics. We have identified multiple novel thermal patterns that better predict coral bleaching on MAR reefs and propose a transferable model to predict the vulnerability of coral reefs to bleaching in the Wider Caribbean region. This model could facilitate the development of emergency responses and conservation strategies

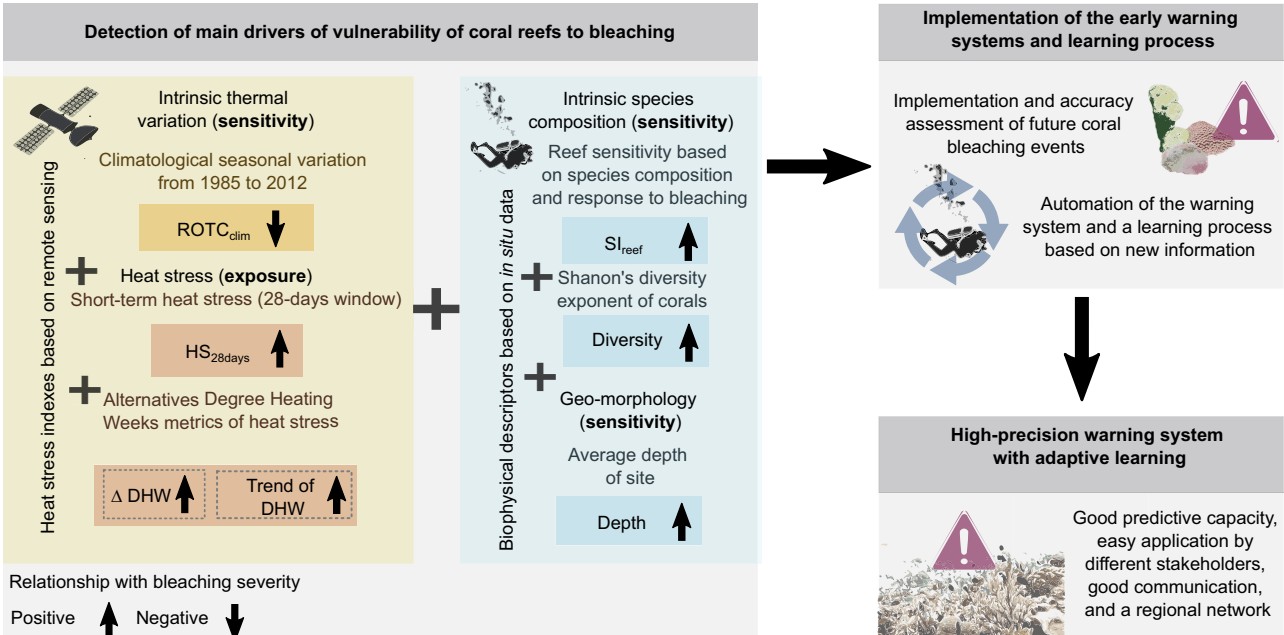

**Fig. 4 | Early warning system for predicting coral bleaching vulnerability in the Mesoamerican Reef.** Conceptual diagram of an early warning system for predicting coral bleaching vulnerability within a regional framework. The left panel identifies key drivers, including intrinsic thermal variation, heat stress metrics from remote sensing data, and biophysical descriptors (species composition, diversity, and site depth) from in situ data. Positive and negative relationships with bleaching severity are indicated by upward and downward arrows, respectively. The right panels describe the implementation process, emphasizing automation, adaptive learning, and the system's ability to improve accuracy through new information.

through an automated early warning system (Fig. 4). The model offers both, theoretical and practical applications, enabling real-time predictions for reefs where data on coral diversity, depth, historical heat stress, and SST data are available. This high-precision early warning system with adaptive learning capacity would greatly benefit coral reef managers and conservation organizations. While there are already global operational warning systems for coral bleaching, validation and refining with long-term in situ observations of heat stress and coral bleaching data collected in different reef regions of the Caribbean and beyond, can be used to refine these global bleaching predictive models into more accurate ecoregional scales. We highlight the need for collaborative coral bleaching monitoring networks along with emergency response plans and data-sharing platforms (such as https://www.healthyreefs.org/ and https://www.agrra.org/). Given the increasing frequency and severity of coral bleaching events, improving these early warning systems for managers and conservation planning efforts is crucial.

## Methods
### Field methods and study area
To assess bleaching severity, the "bar-drop" method was employed to survey a minimum of 150–200 individual coral colonies using a 1 m PVC bar with 5 marks every 25 cm. The bar was haphazardly placed across the reef after 3–4 kick cycles[45]. Corals were identified at the species level and assessed using three bleaching categories: pale colony, partially bleached colony, and whole colony bleached with over 90% of live tissue affected; and one category for non-affected colonies, which were recorded as 'normal'. Sampling was conducted throughout the whole MAR region in three periods: October–November 2015, 2016, and 2017. Sites were selected based on information from previous monitoring programs in the region. Site selection was stratified according to cross-shelf position (e.g., bank reefs, patch reefs, and fringing reefs), the reef zone (e.g., crest and forereef), depth, and wind exposure (e.g., wave exposure). Most of the selected sites had consistent information in other regional databases (e.g., Atlantic Gulf and Rapid Reef Assessment-Healthy Reefs Initiative, protected areas), and we prioritized areas based on the experience and feasibility of the surveys achieved by

local experts. The monitoring was conducted by trained volunteers from various partner institutions of the Healthy Reefs Initiative within Mexico, Belize, Guatemala, and Honduras. Considering the three sampling periods, 266 reef-level samples/observations were obtained: 69 in 2015, 104 in 2016, and 93 in 2017.

### Bleaching severity index
The bleaching severity index (Eq. 1) was adapted from the bleaching and mortality indexes (BMI)[44] and calculated from the sum of the proportion of colonies in each response category, weighting each category according to its ecological impact.

$$Bleaching\ severity = \frac{((c2 + 2 * c3 + 3 * c4))/3}{n} \quad (1)$$

In this equation, '$n$' corresponds to the total number of colonies, and '$c$' represents the number of colonies in each of the categories of concern ($c2$: pale, $c3$: partially bleached, $c4$: whole bleached). Bleaching severity was calculated for each of the reefs and each of the species considering all the colonies of each species.

### Coral sensitivity to bleaching
Five different metrics were calculated to describe the sensitivity of corals to bleach, based on species composition and reef diversity (Table 1). The first step in obtaining these indicators was the selection of the database, including only the colonies identified at the species level.

The reef-level sensitivity ($SI_{reef}$, Eq. 2) was calculated from the sum of the relative abundance of each species (Number of coral colonies; Ncci) multiplied by its bleaching severity value (BSIsp; Supplementary Table 4; Fig. 3), obtaining an expected sensitivity response based on the abundance of the species at each site[3].

$$SI_{reef} = \sum \left( \left( \frac{Ncci}{100} \right) * BSIspi \right) \quad (2)$$

As an approximation to characterize structural complexity, the reef functional index (RFI; Eq. 3) was calculated based on the summation of the abundance (the number of coral colonies; Ncc) multiplied by a functional-coefficient (Fc) of each species for the reef site[71]. The functional coefficient considers multiple morphological and growth characteristics of each coral species present in the region[71]. When species had no functional coefficient, the value available for congeners was used (e.g., *Solesnatrea hyades* was used for *Solenastrea bournoni*). Colonies were not considered when a value for the species could not be assigned (i.e., *Oculina diffusa*).

$$RFI = \sum \left( \left( \frac{Ncci}{100} \right) * Fci \right) \quad (3)$$

We estimated coral species richness and diversity, using N1 of Hill numbers as a metric for diversity (Eq. 4), which is equal to the exponent of Shannon's diversity index[80]. This indicator expresses reef diversity in effective species. The effective species is the number of species in the community in which all species were equally common, this represents the true diversity without considering the less abundant or "rare" species[80]. Under this framework of Hill's numbers, the diversity of a community is measured as the effective number of species in it, which can be understood as the number of species in a virtual, perfectly balanced community, in which all species are equally common, and in which the average relative abundance of the species in the real community is conserved[80]. This diversity indicator considers both the richness or number of species (*s*), as well as the proportion or relative abundance of each species (*pi*).

$$DiversityN_1 = exp \left( -\sum_{i=1}^{s} pi * \ln(pi) \right) \quad (4)$$

As an approximation for the characterization of the ecological gradient based on the composition of coral species, a detrended correspondence analysis (DCA) was calculated[81]. This multidimensional ordination analysis was conducted based on the relative coral species abundance at each site. The position of the sites on the first axis of the DCA was taken as the final metric (DCA1; Supplementary Fig. 3), and this ecological gradient could be related to the variation in coral bleaching severity[3]. Diversity from the Hill N1 and the DCA were calculated using functions available in the vegan[82] package of the R statistics program[83].

### SST data and heat stress metrics

To characterize the effect of different descriptors of heat stress on the response of corals and reefs and the expression of bleaching, 17 metrics were calculated based on the variation of sea surface temperature (SST; Table 1). SST and heat stress metrics were obtained from the CoralTemp database of Coral Reef Watch (https://coralreefwatch.noaa.gov/product/5km/index.php). This database has a resolution of ~5 km and a period from 1985 to the present, with a daily frequency[84]. Additionally, maximum monthly mean (MMM) values were obtained from the same database.

Among these metrics, two represent climatological indicators with a period considered of 1985–2012, these indicators were the MMM and the climatological value of the rate of temperature change (ROTC$_{clim}$). MMM represents the monthly average of the hottest month registered from 1985 to 2012[37], this climatological value comes from the analysis of the climatology of all the years, selecting the value of the hottest month (usually September within the Caribbean coral reefs). This value represents the average temperature for the hottest month between 1985 and 2012 for each satellite pixel. The rate of temperature change (ROTC; termed rate of seasonal warming in Chollett et al.)[9] was calculated for each site, and we (1) calculated the weekly average temperature; (2) identified the maximum; (3) calculated the rate of temperature for the year as the trend for the previous 3 months; (4) in the case of ROTC$_{clim}$ we calculated an average of all rates for that site from 1985 to 2012. Besides the climatological indicators, we considered

aspects of the recent thermal variation, calculating the maximum rate of temperature change observed in the last three months (ROTC$_{84days}$) and the standard deviation of the SST considering 3 months before the sampling (SD$_{84days}$; Table 1).

Furthermore, we calculated novel metrics to characterize the accumulated, acute, and chronic heat stress (Table 1). The indicators generated are based on two main metrics known as Hotspot (HS; Eq. 5) and degree heating weeks (DHW; Eq. 6), which consider thermal anomalies or heat accumulated above the MMM over a certain period[37]. Hotspots (HS) represent daily positive anomalies above the MMM[37]. Heritage DHW quantifies heat stress by summing up positive daily anomalies over 1 °C above the MMM over 84 days (12 weeks), divided by 7 to express values per week[37].

$$HS = \begin{cases} SST_{daily} - MMM, SST_{daily} > MMM \\ 0, SST_{daily,} \leq MMM. \end{cases} \quad (5)$$

$$DHW = \frac{1}{7} \sum_{i=1}^{j=84} (HS_i, \text{ if } HS_i \geq 1\,°C) \quad (6)$$

The acute heat stress metrics calculated were the mean of the HS (positive thermal anomalies greater than the MMM) in the last three days (HS$_{3days}$), and four indicators representing the number of days with anomalies ≥1 or 2 °C within the last 28 (HS1$_{28days}$ and HS2$_{28days}$) and 84 days (HS1$_{84days}$ and HS2$_{84days}$; Table 1). Accumulated heat stress was estimated from the summation of HS in the last 28 (HS$_{28days}$), 84 (HS$_{84days}$), and 555 days (HS$_{555days}$). Additionally, we calculated the metrics of DHW using these same timeframes (28, 84, and 555 days). These metrics of accumulated and acute heat stress have been widely used in several coral bleaching modeling and prediction efforts, being considered commonly the most relevant predictors of coral bleaching[3,15,16,21].

Based on the heritage DHW (calculated using 84 days) we generated two metrics of chronic or long-term exposure patterns. The 'Trend of DHW' was calculated as an indicator of long-term exposure and the interannual trend of heat stress, this metric is the trend of annual maximum DHWs from 1985 to the sampling year, a trend obtained from a generalized least square model[40]. The second metric was the 'ΔDHW', this metric represents the difference between the maximum observed value of DHW in the current event up to the sampling date and the maximum observed value of DHW in the last year (building on Hughes[18]), this metric provides information on the relative magnitude of the heat stress event as a function of the previous year's exposure.

### Statistic data analysis

To identify temporal differences in bleaching patterns between the three years under consideration, a Yuen's test (robust *t*-test) on trimmed means for dependent samples was carried out[85]. The temporal comparison was performed considering only the sites re-sampled in both years in the paired comparisons. Yuen's test was chosen because the values of bleaching severity and bleaching categories in re-sampled sites generally were not homogeneous in variance or normality. The test of normality applied was Shapiro–Wilk. Levene's test was also used to check the homogeneity of the variances. For Yuen's test, a trimmed value of 0.10 was used, eliminating 10% of the outliers on each side of the distribution for a more robust comparison. This test was performed with the "yuend" function available in the "WRS2" library[85] of the R statistics program[83]. The function used estimates of an explanatory measure of effect size, this was realized using a robust heteroscedastic approach for two or more groups[85].

We used GBMs to analyze the relationship between metrics identified as potential drivers (Table 1) and bleaching severity. This statistical analysis based on an assembly of multiple regression trees identifies non-linear relationships, and interactions, which determine with high confidence the relative relevance of the predictive variables[41,42]. GBMs are excellent predictive approximations and are based on a process that facilitates the

identification of the relative importance of all variables with some predictive capacity. This allows the elimination of collinearities and non-relevant or redundant variables, which significantly improves the model and its predictive capacity[41,42]. Based on the GBMs, we evaluated all metrics considered and indicated in Table 1. For this analysis, we assumed the Gaussian distribution of errors based on a visual evaluation of the bleaching severity distribution of the total data. For GBMs we considered a k-fold =10, dividing the total data set in the ten subsets used for the evaluation of the cross-validation and the optimization of the parameters considered in the GBMs[41]. As initial parameters, we chose the recommended (learning rate from 0.01 to 0.001, complexity of regression trees from 3 to 5, and out-of-bag fraction of 0.5), ensuring a minimum of 1000 regression trees[41]. For this procedure, we used the functions available in the gbm[86] and dismo[87] libraries of the R statistics program[83]. In the first global GBM model, all variables were included to identify the relative relevance of the metrics considered and the presence and importance of potential interactions. This global model was simplified and optimized by eliminating the variables that did not contribute to explaining the variation observed in bleaching severity and predicted by the model (Supplementary Fig. 2). In this process of model simplification, special care was taken to eliminate collinearities, selecting only variables with correlation values below 0.6 (Supplementary Fig. 4) and variance inflation factor values below 4[88,89]. The partial dependence plots were made to visualize the relationship between the selected predictive variables and the predicted bleaching severity response. For these graphs, the 95% confidence intervals were calculated from a bootstrap approach with 1000 permutations[43]. We used the H-statistic[90] to evaluate possible interactions among the metrics considered in the GBMs. This algorithm compares the variance of the model's response using two metrics, which separately and combining the partial effect of each metric, allows obtaining a scaled and normalized value that quantifies in the model the strength of the interaction between the two metrics. Finally, the occurrence of spatial autocorrelation was evaluated from a spline correlogram using the functions in the ncf package[91] of the R statistics program (R Core Team, 2017).

## Reporting summary

Further information on research design is available in the Nature Portfolio Reporting Summary linked to this article.

## Data availability

All source data underlying the graphs and charts presented in the main figures of this study are available as Supplementary Material and Supplementary Data 1–2. These data include detailed site-specific metrics on coral bleaching severity, environmental predictors, and species composition used in the analyses. The complete datasets generated during and/or analyzed during the study are available from the corresponding author upon reasonable request.

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

## Acknowledgements

We thank many individuals and partners of the Healthy Reefs Initiative throughout the Mesoamerican Reef for their assistance with conducting field surveys (Supplementary Table 5). Funding for HRIs Bleach Watch Program 2015–2017 was provided by the Summit Foundation, The Oak Foundation, and Code Blue Foundation and supported by in-kind contributions and assistance from over a dozen local partner organizations (Supplementary Table 5). The authors also thank the NOAA Coral Reef Watch program for the availability of the SST data. This paper is part of the Ph.D. program of AIMC in the postgraduate program of Marine Science at CINVESTAV, Unidad Mérida. This program is acknowledged for providing four years of a CONACYT fellowship with grant numbers 340074 and 666908 to support the Ph.D. degree of AIMC and ARS, respectively. Special thanks to the Coastal Biodiversity Resilience to Increasing Extreme Events in Central America (CORESCAM) research project. The scientific results and conclusions, as well as any views or opinions expressed herein, are those of the authors and do not necessarily reflect the views of sponsoring organizations.

## Author contributions

M.M., I.D., M.R., M.S., A.G., and N.C. managed planning and funding for fieldwork, coordinating the coral bleaching monitoring response plan, supported by assistance from A.I.M.C., A.R.S., I.C., and J.E.A.G. A.I.M.C., A.R.S., M.M., I.C., J.E.A.G., and M.E. developed the study concept and analytical framework. A.I.M.C. carried out the statistical analyses and figures with contributions from A.R.S., M.M., I.C., J.E.A.G., S.E., and M.E. A.I.M.C. led the writing with contributions from all authors.

## Competing interests

The authors declare no competing interests.
