## [Peer Review file · Communications Biology]

Underlying drivers of coral reef vulnerability to bleaching in the Mesoamerican Reef

Corresponding Author: Dr Aarón Muñoz-Castillo

Version 0:

Reviewer comments:

Reviewer #1

(Remarks to the Author)

General Comments:

Muñoz-Castillo and colleagues present a comprehensive, novel, and impressive analysis of abiotic and biotic predictors of coral bleaching susceptibility during 2015-2017 across the Mesoamerican Reef. This is an impressive expansion of some of their previous site-specific work and generates some very interesting hypotheses about drivers of coral reef vulnerability to bleaching. While I don't have the statistical expertise to evaluate the analytical approach of the Gradient Boosted Models method, I found the rest of their work comprehensive, rigorous, and robust. The writing and presentation of the results was clear and generally the conclusions were sound (though see below). I do recommend some additional clarification/specifics for how some of their metrics were calculated to aid in interpretation of their data and potential follow-on studies. Even better, would be an electronic repository (github or similar) with scripts/calculations for their metrics and as many raw (or cleaned up) data as possible (see for example the github repository of the manuscript of Sully et al. 2019 Nature Communications; <https://github.com/InstituteForGlobalEcology/Coral-bleaching-a-global-analysis-of-the-past-two-decades>). Overall, I recommend publication of this manuscript following minor revision. Generally, I feel the biggest improvement is strengthening and clarifying some of the interpretations of the results, specifically the significance of the ROTCclim, deltaDHW, and diversity metrics. I've outlined specific comments and suggested additional interpretation in the sections below, in addition to other minor line-by-line comments. The authors are to be commended on an interesting and comprehensive piece of work, and following some minor revision I believe it will be an important contribution to the field. Best regards, Dan Barshis

Specific Comments:

Results and Discussion:

Lines 104-109, I find this sentence confusing. Wouldn't one expect in situ observations of coral bleaching always to predict the spatial variation of bleaching severity (i.e., you're calculating bleaching severity based on your in situ observations). I suggest re-writing this sentence to be more specific and/or remove the part about the in situ observations ... Also suggest replacing the word "excellent" with "strong" or something less qualitative.

Figure 2. Legend, also please specify what the black hash marks on the x-axes of the various panels in d mean, I'm guessing they are the values of the surveyed reef data?

Lines 116-117. The definition of ROTCclim here is different than the one below on lines 126-127. Please expand the definition on lines 126-127 and modify the definition on lines 116-117 to match. Based on my review of the original citation for this metric (Chollett et al 2014), this equates to the following from Chollet: "For all years, weekly average temperatures are calculated, and the maximum of those selected as summer maximum. Then the slope of the line of the three month period prior to that date is calculated (Figure 3f)."

Is that correct? Did you do a similar calculation to detect the summer maximum weekly average and then calculate the slope of the three months prior to that date? Did you use the same summer timepoint for each year (i.e., were the weekly averages for the entire 1985-2012 time period calculated and only a single summer maximum chosen, or was this done within each year)? As this is your top predictor variable, I suggest being very clear about the methodology used, as it is not called ROTCclim in the Chollett et al 2014 paper.

For example "ROTCclim expresses the mean trend of temperature change over three months during the summer" should actually read something more like the following "ROTCclim expresses the mean trend of temperature change over the spring to summer transition (i.e., three months prior to the maximum weekly average temperature)". Also why were the data from 2012-2015 (or 2016,2017) not used? How might this have changed the results?

Line 138, note that Chollett et al. 2014 do not use the term ROTCclim, suggest modifying to "ROTCclim (termed rate of seasonal warming in Chollett et al. 2014)"

Lines 137-146, I find the second half of this paragraph a bit confusing. The supplementary information from Chollett et al outline two potential scenarios, one where a fast rate of seasonal warming triggers faster loss in coral pigmentation and symbiont content and a more fragile phenotype, and the other where steady increases in temperature from spring to summer provide enough time for corals to fully develop the summer phenotype and acclimate to summer temperatures creating a more robust phenotype. Your data would support the latter so I would suggest rewriting this paragraph to draw a more solid conclusion. Here you show the strongest predictor of reduced bleaching severity is a faster rate of change from spring to summer over the historical climatology. Couldn't this environmental pattern have then selected for or created acclimated populations that benefit from this faster rate of change? I understand the data are not definitive, but they are also clear in supporting a beneficial influence of higher ROTCclim so it's okay to say that and discuss potential mechanisms to support that conclusion versus go into detail about why it might not be true.

Lines 147-164, I wonder whether there is any relevance of some of the recent "marine heatwave" literature or at least a discussion of how recent bleaching events may have been more intense and faster-paced than prior events. One could imagine that DHWs may be more relevant to older bleaching event trajectories while the HS28days as a stronger predictor of the data herein signifying a shift in the onset and rate of increase in more recent bleaching events (eventually leading to a marine heatwave trajectory).

Not sure if relevant but here are some recent marine heatwave papers:

Leggat et al 2019 *CurBiol* Rapid Coral Decay Is Associated with Marine Heatwave Mortality Events on Reefs

Fordyce, A. J., T. D. Ainsworth, S. F. Heron, and W. Leggat. 2019. Marine Heatwave hotspots in coral reef environments: Physical drivers, ecophysiological outcomes, and impact upon structural complexity. *Front. Mar. Sci.* 6: 498

Hobday, A. J., and others. 2016. A hierarchical approach to defining marine heatwaves. *Prog. Oceanogr.* 141: 227–238

Line 173-174, see comment below regarding the specific equation for deltaDHW determination. Based on Table 1 it sounds opposite to what is specified here. Regardless, I suggest expanding the discussion of the meaning of higher deltaDHW a bit more, as the Hughes et al 2019 paper suggests that sites that bleached in 2016 bleached less in 2017 so counter to what is discussed here. Based on the deltaDHW panel in figure 2d it seems as though there is not much effect until a delta DHW > ~6, which I would interpret as more severe bleaching in sites where the bleaching event in the sampling year was more severe than the previous year. This would agree with the longer-term trend of DHW but over a much shorter timescale. A deltaDHW closer to zero or negative would mean the prior year was as severe or more severe than the sampling year (based on the description of the calculation in the text, not table 1).

Lines 188-209, one additional explanation could be that for a species like *Acropora palmata*, the more sensitive individuals have been wiped out, thus only more resilient individuals still exist. At least some mention of potential selective changes over the past few decades seems relevant to add to this section. See "Guest et al 2012 *PlosOne* Contrasting patterns of coral bleaching susceptibility in 2010 suggest an adaptive response to thermal stress" and similar references.

Lines 220, add "Smith et al 2016 *GCB* Caribbean mesophotic coral ecosystems are unlikely climate change refugia" as an example of increasing bleaching susceptibility with depth as well.

Lines 223-239, couldn't another explanation be that more diverse reefs have a higher chance of having bleaching sensitive species in abundance? I.e., the more species you have the more likely it is some of those species are sensitive?

Figure 4. Doesn't delta DHW have a positive association with bleaching severity (i.e., higher delta DHW means higher bleaching severity)? Again, see comments above and below to clarify the specific calculation used for this metric.

Methods:

Line 287 change to "sensitivity of corals to bleach[ing]"

Lines 316-317 see comments above concerning the need for a more detailed explanation of the ROTCclim variable, specifically whether the three month period was the same for each year or calculated for each year individually based on the average weekly maximum temperature as in Chollett et al. 2014

Table 1 DeltaDHW field. Suggest additional clarification here as well. Particularly, which value was subtracted from which. In the table, it sounds as if the formula was this: $\text{maxDHW}_{\text{lastyear}} - \text{maxDHW}_{\text{currentevent}}$ but in the text it sounds like it's the opposite $\text{maxDHW}_{\text{currentevent}} - \text{maxDHW}_{\text{lastyear}}$. It makes a big difference to the interpretation. Also the Hughes et al 2019 reference does not use the delta DHW terminology so is misleading to cite as the basis for this indicator using that name, perhaps a "sensu" or "building on" preface to the Hughes citation is appropriate to add.

Reviewer #2

(Remarks to the Author)

This is beautiful paper – really well illustrated and discussed, and covering a complex field concisely. The results are very important in the literature, showing a well-constructed analysis of multiple drivers/factors affecting coral bleaching. It provides a very important next-step in this field of research.

There are some problems that I address in more detailed comments – these do need to be addressed for the paper to not fall into some of the traps that are common in this field (particularly, having the ‘final say’ on what controls coral bleaching and that results must be relevant everywhere).

I certainly recommend for publication with these points addressed.

There are no line numbers, so I copy out text fragments to help locate my comments.

Abstract

The paper presents good discussions on the conflicting results that are in the literature (e.g. deep reefs being a refuge vs. being more vulnerable) ... it all depends, but falls a bit into the same trap proposing their results can be “transferable to other regions” ... the statement in the abstract is good, “tools to be explored by coral research and conservation programs”, but in some places in the text falls into the trap of claiming to provide a predictive algorithm that can be applied elsewhere. See more detailed comments.

“Here we demonstrate a regional warning system” ... no, this does not demonstrate a warning system. It analyses factors associated with variation that can be used in a warning system. The study looks at factors associated in an event(s) that happened – how well this will perform in a predictive/warning system needs to be developed and tested. So the orientation of the primary claims of the paper need to be shifted to reflect this. The last sentence in the intro is correct that there is potential for this use, not demonstration of it here “.. with a high potential for use as an early warning system for the Mesoamerican Reefs transferable to other regions”. And see comment below, what is ‘regional’ – the spatial scope of the study must be stated in the abstract.

“multiple drivers of coral bleaching” – some of these are ‘drivers’, but some are factors affecting the bleaching response. Need to be more explicit about these roles.

The last sentence in the abstract suggests using this in other regions, which is great, but the abstract (nor the paper) is not explicit about what a ‘region’ is – the Caribbean, the MAR? The abstract must state the scale of this analysis, ie. the MAR.

Main text

“We analysed remote sensing data and 266 in situ observations recorded during the seasonal “bleaching window” ... see later comment on ‘samples’ vs. observations. The observation that is analysed is each coral colony record, not each reef sample ... If I’ve got this wrong, perhaps it’s the % bleaching levels and BSI that are analysed at reef level, then this needs to be made clearer. See comment below.

Species with thin tissues and flat-extrapolate colony morphologies – is “extrapolate” a correct word, or should it be ‘explanate’

“Our findings do not question this conclusion but highlight the need for a better understanding of the cellular mechanisms behind the physiological disturbance of this symbiotic association and all the biological processes that can confer stress resistance at the symbiont, host, and holobiont/microbiome levels.” – or is it just that there are a great many factors influencing these processes and its more important to base analysis on observations rather than theory/extrapolation from elsewhere and different times/events?

“Thus, global bleaching predictive models can be refined on regional scales when regional data such as these are available” – again, be explicit about what is meant by “regional” – the Caribbean, part of the Caribbean, the MAR? The implications for developing a warning system and its spatial reach are very different for these.

Methods

“Considering the three sampling periods, 266 observations were obtained at the reef level, for 69 sites in 2015; 104 sites in 2016; and 93 sites in 2017” – this is confusing – an ‘observation’ is more strictly each colony observed/documentated. Is this better to state as ‘samples’, or reorganize as “266 reef level samples/observations were obtained, 69 in 2015”

These statements are only understandable to people who know the details of these, consider explaining for a broader audience: “N1 of Hill numbers” and “The effective species is the number of species in the community in which all species were equally common, this represents the true diversity without considering the less abundant or “rare” species”

Fig 4 – the integrated diagram is good, but the proximity of the ‘+’ signs within the yellow box and the one outside is confusing. Perhaps move the inner ones to the left, mirroring the blue box.

Supplementary Figure 3. Caption – should the second sentence start with “We”?

(Remarks to the Author)

The paper is a description of the bleaching surveys undertaken in the MesoAmerican Reefs from 2015 to 2017. The authors describe the bleaching intensity of the taxa in space and time and evaluate a number of thermal and depth metrics in predicting the response levels. This represents a large and well coordinated program of study that is unique in the Caribbean. The original part of this study is the use of the machine learning algorithm to evaluate the potential influence of various thermal metrics. The produces a good fit to the data, which is a good outcome and considerably improves on previous efforts (not cited) to do this on a large scale. Below I list a number of the weaknesses of the paper that may be rectified with a major revision.

The paper is descriptive and not really driven by any specific or alternative or competing stress theories or hypothesis. It has a data-mining approach that leads to a weak organization and descriptive conclusions. Additionally, the organization is weak in terms of focusing on the original findings and the main points that are contributing to science, as opposed to a description of what occurred in MBR reefs during this period. The authors need to rethink what should be in the main and supplementary text and elements. Many of the variables are what are typically evaluated by NOAA in terms of their metrics of stress. But, the literature has moved on from these early recommendations of NOAA and is examining more variables and testing more theories of stress. The paper therefore feels dated as the citation is not current and there is little or no connection to a number of important recent studies of bleaching and coral mortality. The authors need to search the literature since 2016 and incorporate these newer papers into their approach. Many recent studies are finding variables that are important that are not examined here. Many of the studied variables here are not well described and confusing and only seldom are equations given. Table 1 is a good place to organize more effectively. This makes it hard to confirm the methods and interpret the data, which leads to weak support for the paper as science. What effect might the many issues of variables studied here and elsewhere have a results, interpretation, and comparison with the larger literature?

I have limited my detailed comments as I think these larger focus, context, and compositional issues will need to be addressed before a decision can be made.

Abstract

L20 – not more frequent but longer duration – see Skirving et al. 2019

L26 – not sure I understand if these are in the year of the bleaching or historically. What is heat stress here, needs a definition? Studies like this often distinguish between historical and current metrics. See Sully et al. 2020.

One gets the feeling these reefs were already exposed to thermal stress and therefore deep reefs are now being affected. Maybe something about the contextual location and time period of this study would be useful for context. The abstract is somewhat mysterious because it is so short.

I also wonder if one could actually get these data prior to a bleaching event. In seems this is post bleaching analysis with data that may not have been available prior to the bleaching and therefore theoretical and not applicable to actually predicting bleaching. Can this be discussed later in the text?

Introduction

A number of recent studies are suggesting geographic position or some regional differences in bleaching. Wondering why this is not mentioned as it can be a dominant factor? Perhaps this is not so important for MBE but was found for some ecoregions where coral cover and bleaching were evaluated (Verccammen et al. 2019; McClanahan et al. 2020).

L73- given there are no methods until here in the reading, can a bit more be said about what the BSI is? Otherwise, the rest of the text is hard to interpret.

L76- what is an observation? I assume it is some sample of many colonies but this seems important to make clear early.

L84 – not sure what spatial footprint means?

Figure 1 has so many panels that it is quite confusing to interpret. The problem with Nature paper is putting multiple figures on a single figure. Is A) summarized for all years? B) –what is index? Not really sure what C) is even after reading the methods. Seems like a supplementary figure. How important is all of these panels for the main conclusions of the paper? Maybe less information here and more in the supplement would work better. If the main point of this figure is that 2017 was the worst year, this is too much information. Could a simpler graph that shows the severity in space and time in 2 panels all that is needed to make the main point? Then most of these descriptive details can go to the supplement.

Does figure 2 really require so many abbreviations? It just makes it hard to follow if a reader is not familiar with these abbreviations. At least bring the table forward so readers can see them, their definition, and maybe equations.

The negative relationship with ROTC is provocative finding given all models to date use it to predict disasters for reefs, i.e. Hoegh Guldberg 1999, etc. This seems important to spend some text on, as the future models are likely to be quite poor predictors if this is more widespread.

P125 – this discussion of summer and winter phenotypes seems to assume this is true for all corals. How well studies and broad is this seasonal pattern concept? This seems speculative and not really critical for the paper.

L187 – why not use historical data as a predictor? I think this was done in the Sully et al. 2019 Nat. Comm. paper.

One gets the feeling that some taxa, like Acropora, do not bleach but die. This is one option a maybe this bleach/no bleach vs die/no die needs some consideration. Can you use the mortality data to evaluate this for the studied taxa? Perhaps the BSI is not good at distinguishing bleaching versus dying?

The discussion section would benefit from stronger English composition. It rambles and is not highly organized around key conclusions, caveats, and context.

Methods

L272-275 – this site selection text is not very clear as per the decisions, only that they had been done in the past. Please explain how or why they were originally selected. Is there any experimental design in site selection?

L292- I believe this RFI values needs an equation to be better understood.

P309 – I find most of these metrics not well explained and wonder if equations would help. For example, is this a new type of MMM? It is not that clear as described and therefore confusing as previous MMM are the 3 hottest months of the year over time. How can one take an average of a hottest month over time? This would be only 1 month. I think there is some poor writing going on here but this text has to be clear to be evaluated.

I like table 1 and wonder if it could be in the main text with a column for the equations and maybe citations of there they were first used? What the ranges of values are and what are the units, etc. This would help greatly with interpretation. See Nature Climate Change 9:845-851 for a previous example published in the Nature series.

Table 1 is many heat stress metrics but not much consideration of the other modifying variables including light, water quality, etc. Is there some reason for this focus? It seems there are not really any hypotheses driving this paper but rather a bit of a data mining procedure. Can the data mining aspect be reduced?

P325 – best to start a paragraph saying why you are doing this text between years? It seems tacked in otherwise.

GBM sounds a lot like a BRT. Are they the same?

L350 – Best to give short results of what the final variables selected were and why. Why not put results in table 1?

There are many relevant recent papers that are not cited, even ones directly related to the Caribbean let alone the larger reef locations.

Welle, P. D., Small, M. J., Doney, S. C., & Azevedo, I. L. (2017). Estimating the effect of multiple environmental stressors on coral bleaching and mortality. PLoS One, 12(5), e0175018.

Version 1:

Reviewer comments:

Reviewer #4

(Remarks to the Author)

To the editor and authors:

I have read the revised manuscript and the rebuttal document. In my opinion, the authors have responded thoughtfully and fully to all the major suggestions made by the three reviewers. I do not have any concerns with their response, I see no conceptual flaws, the conclusions are original, and the paper contributes new knowledge that will appeal to a broad science-to-management audience involved in coral reef issues. Some of the prose, especially in the results / discussion section, remains hard to read and is repetitive in places. To remedy some of this, I here make a few suggestions:

Line 158: The "rate of temperature change from 1985 to 2012" is a misleading way of stating what this variable is when it is defined differently in the next sentence. Recommend change to something like: "We found the climatological seasonal-warming rate (sensu Chollet et al. 2014) from 1985 to 2012 (ROTCClim) to be the main driver of the severity of coral bleaching (Fig. 2a). ROTCClim expresses the mean trend of temperature change over the spring-to-summer transition (i.e., during the three months before the maximum weekly average temperature) from 1985-2012. (and delete "this metric has also been termed the rate of seasonal warming9).

Line 162, delete the repetitive sentence "We found that higher..."

Line 197, replace "in this sense" with "; thus, it is..."

Line 201, delete the repetitive sentence "These differences can be addressed..." You just stated that above.

Line 214, “the difference between two consecutive years” is ambiguous, change to “the difference between most recent and previous year.” Then delete repetitive sentence in line 216: “These were sites with...”

Lines 216 to 225, this section on delta DHW is still confusingly written and needs refinement. State more clearly and simply what Hughes et al (2019) showed. Delete sentence in line 225 “This result is probably...”

Line 244, perhaps the circular nature of this conclusion should be acknowledged. If I understand correctly, the bleaching response data were used to define SI for each species, and then related back to observed bleaching severity, is this correct? Or were different years’ data used to define SI?

Line 247 and 248, these two sentences seem to directly contradict each other. Perhaps fix by saying “does not agree with SOME previous findings.”

Line 273, delete repetitive sentence “Higher bleaching severity...”

Line 275, replace “These” with “Thus, our results...” and line 277, connect the two sentences with “...heat and light attenuation, and agree...”

Line 301 delete “emphasizes the importance of understanding the” and use more direct language like “In conclusion, our study revealed key drivers...”

Line 305 to 307, this is not a complete sentence. Rephrase. I think what is meant is that the model has both theoretical and practical application for reefs where diversity, depth, historical heat stress, and real-time SST data are available.

Figure 1 – Recommend inserting scale bar and north-pointing arrow.

Figure 2 – Caption, line 144 in a) delete “the” and “was”; b) what are the “raw data” shown – are they bleaching severity? If so, what are units, percent? Add units to x-axis “Distance (km)” on the figure and in caption, add distance from what. Is it distance along the MAR going north to south? South to north?

Lastly, make sure to update the supplementary material document to reflect suggestions made by the three reviewers and me, especially with regards to variable definitions and explanations. Also, clean up this document to have better resolution figures, consistent numbers of significant digits in tables, and add a cover page with title, authors, etc.

The images or other third party material in this Peer Review File are included in the article’s Creative Commons license, unless indicated otherwise in a credit line to the material. If material is not included in the article’s Creative Commons license and your intended use is not permitted by statutory regulation or exceeds the permitted use, you will need to obtain permission directly from the copyright holder.

Point-by-point response to comments on "Underlying drivers of coral reef vulnerability to bleaching in the Mesoamerican Reef"

Apr 2024

We would like to thank the Reviewers and the Editor for their constructive criticism and special attention during the review. This review allowed us to improve the grammar of some specific sections of the manuscript, and to clarify the robustness and relevance of the analysis.

We provide a point-by-point response to the comments made in the manuscript. Reviewers' comments are in black; authors' responses are in dark blue text. Changes in the manuscript have been highlighted in the revised version. The main additions/editions to the document have been included in this archive.

Referee expertise:

Referee #1: Coral ecophysiology climate change

Referee #2: Marine ecology, coral reefs

Referee #3: Marine conservation, coral ecology

Reviewers' comments:

Reviewer #1 (Remarks to the Author):

General Comments:

Muñiz-Castillo and colleagues present a comprehensive, novel, and impressive analysis of abiotic and biotic predictors of coral bleaching susceptibility during 2015-2017 across the Mesoamerican Reef. This is an impressive expansion of some of their previous site-specific work and generates some very interesting hypotheses about drivers of coral reef vulnerability to bleaching. While I don't have the statistical expertise to evaluate the analytical approach of the Gradient Boosted Models method, I found the rest of their work comprehensive, rigorous, and robust. The writing and presentation of the results was clear and generally the conclusions were sound (though see below). I do recommend some additional clarification/specifics for how some of their metrics were calculated to aid in interpretation of their data and potential follow-on studies. Even better, would be an electronic repository (github or similar) with scripts/calculations for their metrics and as many raw (or cleaned up) data as possible (see for example the github repository of the manuscript of Sully et al. 2019 Nature Communications; <https://github.com/InstituteForGlobalEcology/Coral-bleaching-a-global-analysis-of-the-past-two-decades>). Overall, I recommend publication of this manuscript following minor revision. Generally, I feel the biggest improvement is strengthening and clarifying some of the interpretations of the results, specifically the significance of the ROTCclim, deltaDHW, and diversity metrics. I've outlined specific comments and

suggested additional interpretation in the sections below, in addition to other minor line-by-line comments. The authors are to be commended on an interesting and comprehensive piece of work, and following some minor revision I believe it will be an important contribution to the field.

Best regards,
Dan Barshis

Thank you for your thorough comments. We apologize for the delay in the response process. We have carefully considered your suggestions and included better explanations by adding Table 1, sections a and b. Regarding your suggestion for an electronic repository, we can share the data upon request to the authors; however, we are unable to share the raw data as we are collaborating with other partners who have ownership over portions of the dataset.

Specific Comments:

Results and Discussion:

Lines 104-109, I find this sentence confusing. Wouldn't one expect in situ observations of coral bleaching always to predict the spatial variation of bleaching severity (i.e., you're calculating bleaching severity based on your in situ observations). I suggest re-writing this sentence to be more specific and/or remove the part about the in situ observations ... Also suggest replacing the word "excellent" with "strong" or something less qualitative.

Thanks for the suggestion. We changed the sentence in lines 120 to 122:

“We found that key heat stress exposure metrics in combination with intrinsic factors such as climatological thermal variability, and reef sensitivity based on species composition, depth, and coral diversity are strong predictors of coral bleaching severity (Table 1; Fig. 2).”

Figure 2. Legend, also please specify what the black hash marks on the x-axes of the various panels in d mean, I'm guessing they are the values of the surveyed reef data?

Thanks for the suggestion. You are correct, we included the definition in the Figure title lines 147 and 148:

“Black hash marks on the x-axes represent the values of the surveyed reef data.”

Lines 116-117. The definition of ROTCclim here is different than the one below on lines 126-127. Please expand the definition on lines 126-127 and modify the definition on lines 116-117 to match. Based on my review of the original citation for this metric (Chollett et al 2014), this equates to the following from Chollet: "For all years, weekly average temperatures are calculated, and the maximum of those selected as summer maximum. Then the slope of the line of the three month period prior to that date is calculated (Figure 3f)."

Is that correct? Did you do a similar calculation to detect the summer maximum weekly average and then calculate the slope of the three months prior to that date? Did you use the same summer timepoint for each year (i.e., were the weekly averages for the entire 1985-2012 time period calculated and only a single summer maximum chosen, or was this done within each year)? As this is your top predictor variable, I suggest being very clear about the methodology used, as it is not called ROTCclim in the Chollett et al 2014 paper. For example "ROTCclim expresses the mean trend of temperature change over three

months during the summer" should actually read something more like the following "ROTC_{clim} expresses the mean trend of temperature change over the spring to summer transition (i.e., three months prior to the maximum weekly average temperature)". Also why were the data from 2012-2015 (or 2016,2017) not used? How might this have changed the results?}

Thank you for your detailed inquiry regarding the ROTC_{clim} metric. We appreciate your thorough examination of our methodology. To clarify, we did indeed perform a similar calculation to detect the summer maximum weekly average temperature and then calculated the slope of the three months before that date for each site and year. Contrary to what was initially stated, the correct definition of ROTC_{clim} should express the mean trend of temperature change over the spring-to-summer transition, encompassing the three months before the maximum weekly average temperature, as outlined in Chollett et al. (2014).

So, for each site and year we (1) calculated the weekly average temperature; (2) identified the maximum; (3) calculated the rate of temperature for the year as the trend for the previous 3 months; (4) calculated an average of all rates for that site.

We apologize for any confusion arising from the discrepancy in our initial definition and have now amended it to align with the accurate methodology described.

Additionally, regarding the exclusion of data from 2013 onwards, we used the average of the trend for all years to ensure consistency in our analysis with other metrics like the Maximum Monthly Mean. So, in the case of the metrics based on climatology, we used the period 1985 to 2012 in the case of all the metrics.

We included some of the modifications regarding this metric:

Lines 131 to 133: "Our results showed that the climatic thermal variation represented by the ROTC_{clim} indicator (termed rate of seasonal warming in Chollett et al. 2014⁹) is a thermal metric that considerably influences the severity of bleaching."

Table 1: "The Rate of Temperature Change (ROTC) reflects the trend in temperature change over 84 weeks during summer⁹. ROTC_{clim} is the average annual ROTC values for the period 1985-2012 (termed rate of seasonal warming in Chollett et al. 2014)⁹."

Lines 148 to 150 in Fig 2. "ROTC_{clim} expresses the mean trend of temperature change during the spring-summer transition (i.e., three months before the maximum weekly average temperature) along the period 1985-2012 also termed rate of seasonal warming⁹."

Lines 158 to 161: "We found the climatological rate of temperature change from 1985 to 2012 (ROTC_{clim}) to be the main driver of the severity of coral bleaching (Fig. 2a). ROTC_{clim} expresses the mean trend of temperature change over the spring-to-summer transition (i.e., three months before the maximum weekly average temperature) along the period 1985-2012, this metric has also been termed the rate of seasonal warming⁹."

Lines 389 to 393: "The rate of temperature change (ROTC; termed rate of seasonal warming in Chollett et al. 2014)⁹ was calculated for each site, and we (1) calculated the weekly average temperature; (2) identified the maximum; (3) calculated the rate of temperature for the year as the trend for the previous 3 months; (4) in the case of ROTC_{clim} we calculated an average of all rates for that site from 1985-2012."

Line 138, note that Chollett et al. 2014 do not use the term ROTC_{clim}, suggest modifying to "ROTC_{clim} (termed rate of seasonal warming in Chollett et al. 2014)"

Thank you for the suggestion, we included that change.

Lines 131 to 133: "Our results showed that the climatic thermal variation represented by the ROTC_{clim} indicator (termed rate of seasonal warming in Chollett et al. 2014⁹) is a thermal metric that considerably influences the severity of bleaching."

Lines 137-146, I find the second half of this paragraph a bit confusing. The supplementary information from Chollett et al outline two potential scenarios, one where a fast rate of seasonal warming triggers faster loss in coral pigmentation and symbiont content and a more fragile phenotype, and the other where steady increases in temperature from spring to summer provide enough time for corals to fully develop the summer phenotype and acclimate to summer temperatures creating a more robust phenotype. Your data would support the latter so I would suggest rewriting this paragraph to draw a more solid conclusion. Here you show the strongest predictor of reduced bleaching severity is a faster rate of change from spring to summer over the historical climatology. Couldn't this environmental pattern have then selected for or created acclimated populations that benefit from this faster rate of change? I understand the data are not definitive, but they are also clear in supporting a beneficial influence of higher ROTC_{clim} so it's okay to say that and discuss potential mechanisms to support that conclusion versus go into detail about why it might not be true.

We acknowledge that our data suggest a beneficial influence of higher ROTC_{clim} on coral resistance to bleaching, potentially indicating the presence of acclimated populations benefiting from rapid seasonal changes. We have revised the paragraph to emphasize this conclusion more solidly, highlighting the role of environmental seasonality in shaping coral resilience to bleaching events. However, we agree that further research is needed to elucidate the precise mechanisms underlying this phenomenon and its implications for coral populations globally.

Lines 158 to 185: We found the climatological rate of temperature change from 1985 to 2012 (ROTC_{clim}) to be the main driver of the severity of coral bleaching (Fig. 2a). ROTC_{clim} expresses the mean trend of temperature change over the spring-to-summer transition (i.e., three months before the maximum weekly average temperature) along the period 1985-2012, this metric has also been termed the rate of seasonal warming⁹. The reefs with the highest ROTC_{clim} had the lowest bleaching severity (Fig. 2d). We found that higher climatological rates of temperature change in summer could be a key contributor to increasing coral resistance to bleaching. Our findings suggest that corals present in reefs with a thermal history characterized by higher and more rapid seasonal change were more resistant during the bleaching event. Environmental seasonality is particularly important for corals, as seasonal variation in irradiance and temperature directly impacts the rate of photosynthetic carbon fixation. In contrast, temperature changes also induce important adjustments in the heterotrophic metabolism. Corals acclimated to areas with moderate to high thermal variations are accustomed to physiological variations and adaptations which may assist their ability to acclimate to thermal stress-associated bleaching events^{16,17,22,30,57-59}. Several Caribbean corals have demonstrated an ability to express two different coral holobiont phenotypes, winter, and summer, with contrasting susceptibility to bleaching under similar heat stress exposure³⁰. The rate of temperature change in spring to early summer may determine when the summer coral phenotype is fully achieved, and therefore when the coral is prepared to cope with heat stress³⁰. This implies that a more robust response could be expected when accumulated heat stress coincides with the

complete expression of the summer phenotype³⁰. However, when the heat stress event occurs before the complete expression of the summer coral phenotype, or if moderate or acute heat stress is prolonged for too long, different physiological responses can be expected³⁰. This could be a physiological mechanism that could explain the results of our work. Still, more research is needed to explain the contribution of ROTC_{clim} to coral bleaching during heat stress events. This is of relevance as it could represent an emerging pattern of potential resilient populations, which may modify the perception we have regarding previous existing models^{9,17,60}, especially if this is an emerging pattern in all reefs worldwide. In our results, the ROTC_{clim} represents a climatological thermal variability metric without collinearity with most of the indicators of heat stress. It is important to mention that high rates of temperature change do not necessarily result in high exposure to heat stress, and sites with high rates of temperature change but low heat stress could be considered as potential refugia^{9,16,57,61}.

Lines 147-164, I wonder whether there is any relevance of some of the recent "marine heatwave" literature or at least a discussion of how recent bleaching events may have been more intense and faster-paced than prior events. One could imagine that DHWs may be more relevant to older bleaching event trajectories while the HS_{28days} as a stronger predictor of the data herein signifying a shift in the onset and rate of increase in more recent bleaching events (eventually leading to a marine heatwave trajectory).

Thank you for bringing up the potential relevance of recent "marine heatwave" literature and the intensity of recent bleaching events. We appreciate your suggestion and have reviewed the papers you mentioned. However, we did not find conclusive evidence regarding the relationship between the shift in the onset of bleaching events and the HS_{28days} metric. It's possible that we may not fully understand the reviewer's suggestion.

Not sure if relevant but here are some recent marine heatwave papers:
Leggat et al 2019 CurBiol Rapid Coral Decay Is Associated with Marine Heatwave Mortality Events on Reefs

The study suggests that coral bleaching events will likely have severe consequences for reef structures shortly due to increasing sea surface temperatures. This highlights the urgent need to understand the implications of marine heatwaves on coral reefs globally. However, we did not find it relevant as a reference for our work.

Fordyce, A. J., T. D. Ainsworth, S. F. Heron, and W. Leggat. 2019. Marine Heatwave hotspots in coral reef environments: Physical drivers, ecophysiological outcomes, and impact upon structural complexity. *Front. Mar. Sci.* 6: 498

We included this reference in different parts of the manuscript, but we did not find any evidence regarding the use of metrics with a shorter temporal window of accumulated heat stress.

Hobday, A. J., and others. 2016. A hierarchical approach to defining marine heatwaves. *Prog. Oceanogr.* 141: 227–238

We did not include this reference because it is not relevant to our work.

Line 173-174, see comment below regarding the specific equation for deltaDHW determination. Based on Table 1 it sounds opposite to what is specified here. Regardless, I suggest expanding the discussion of the meaning of higher deltaDHW a bit more, as the Hughes et al 2019 paper suggests that sites that bleached in 2016 bleached less in 2017

so counter to what is discussed here. Based on the deltaDHW panel in figure 2d it seems as though there is not much effect until a delta DHW > ~6, which I would interpret as more severe bleaching in sites where the bleaching event in the sampling year was more severe than the previous year. This would agree with the longer-term trend of DHW but over a much shorter timescale. A deltaDHW closer to zero or negative would mean the prior year was as severe or more severe than the sampling year (based on the description of the calculation in the text, not table 1).

Thank you for your insightful comment on the Δ DHW equation. We've reviewed Table 1 and the text, acknowledging potential confusion. Δ DHW represents the difference in DHW between the current bleaching event and the previous year, as outlined by Hughes et al. (2019). Our findings show that higher Δ DHW values correspond to more severe bleaching, especially when exceeding 6 °C-weeks, aligning with recent heat stress history's role in predicting bleaching severity, as noted by Hughes et al. (2019).

Table 1

Difference between the maximum observed value of DHW in the current event up to the sampling date and the maximum observed value of DHW in the last year (building on Hughes 2019¹⁹).

Lines 213 to 237: In addition, we observed that the recent heat stress history, represented by the difference between two consecutive years in the maximum DHW (Δ DHW)¹⁹, was an important metric to predict bleaching severity. The highest severity of bleaching was observed in sites with Δ DHW values between 4 to 8 °C-weeks (Fig. 2d). These were sites with a greater impact in the year sampled than in the previous year. High values of Δ DHW express a much higher exposure in the recent event than in the previous year, which was reflected in a considerable impact on the severity of bleaching, especially from values above 6 °C weeks. These results are consistent with the findings of Hughes (2019)¹⁹, who found that bleaching was more severe in reefs with high recent exposure to heat stress and low exposure in the past event or in contrast on reefs with high heat stress in the previous year low bleaching was observed in the subsequent event even when similar heat stress is present. This response to previous events can be generated by various biological and ecological responses including acclimatization, adaptation, and a shift in species composition. Δ DHW expresses the potential physiological or ecological memory that corals may have recorded, induced by past exposure to heat stress^{19,20}. This result is probably related to the negative impact caused by the previous bleaching event. For example, after a site's first significant bleaching event, the most sensitive species would be reduced in relative abundance, leaving a higher abundance of species that can better resist the next heat stress event^{19,70}. In addition, at the level of an individual coral, it has also been documented that high previous exposure to heat stress can make some coral species more resistant to the next heat stress event^{5,66}. However, the ability of a coral to build resistance will depend on the metabolic costs incurred during the stress event and on its ability to fully recover before the next heat stress event occurs. Thus, for a certain range of stress, some corals can have a negative response to consecutive events, and they will not achieve acclimatization^{28,30,66,68}. Considering the existing evidence and the results obtained in this analysis, we conclude that not only the current thermal regime but also the history of heat stress of a particular reef needs to be considered to predict the risk of bleaching during a new event. In summary, the long-term, and recent history of exposure to heat stress is fundamental for the prediction of coral bleaching during long-lasting events.

Lines 188-209, one additional explanation could be that for a species like *Acropora palmata*, the more sensitive individuals have been wiped out, thus only more resilient individuals still exist. At least some mention of potential selective changes over the past few decades seems relevant to add to this section. See "Guest et al 2012 PlosOne

Contrasting patterns of coral bleaching susceptibility in 2010 suggest an adaptive response to thermal stress" and similar references.

Thanks for the comment we included this suggestion and the reference in the corresponding text.

Lines 249 to 255: Species with thin tissues and plated colony morphologies such as those of the genus *Agaricia*, were, as expected, among the most sensitive, while the branching *Acropora palmata* showed the lowest bleaching severity within the most abundant species (Fig. 3). It is important to highlight that some species, especially in the genus *Acropora*, have been exposed to constant selective pressures and a decrease in populations during the last decades, therefore the remaining colonies may be more resistant, as the bleaching observed in these taxa was minor^{73,74}.

Lines 220, add "Smith et al 2016 GCB Caribbean mesophotic coral ecosystems are unlikely climate change refugia" as an example of increasing bleaching susceptibility with depth as well.

Thanks for the suggestion we included the reference in the sentence in lines 273 and 274: "Our results agree with other studies that have found no clear depth refuge for coral species^{34,77-80}."

Lines 223-239, couldn't another explanation be that more diverse reefs have a higher chance of having bleaching sensitive species in abundance? I.e., the more species you have the more likely it is some of those species are sensitive?

Thanks for the suggestion, we incorporate some lines about this in the manuscript.

Lines 289 to 299: This suggests that coral diversity does not necessarily confer protection at the regional scale and is more likely to be an attribute associated with the prior disturbance history. We observed that the severity of bleaching was high in reefs with low diversity but dominated by sensitive species such as *Agaricia tenuifolia* and *Porites porites*, whereas low diversity reefs dominated by more resistant species such as *Porites astreoides* were among the least affected (Supplementary Figure 3). Also, more diverse reefs could have more abundance of different sensitive species. Our results then stress the relevance of multiple metrics and the integration of different approaches (i.e., remote sensing, coral physiology, and ecological surveys) to fully understand the range of coral responses to heat stress and the risk of bleaching on a particular reef. The understanding of why shallower reefs with lower diversity could be less vulnerable to bleaching, or how they handle high environmental variability or form more resistant coral communities after surviving greater past disturbances (ecological filters), requires a complex and integrative approach.

Figure 4. Doesn't delta DHW have a positive association with bleaching severity (i.e., higher delta DHW means higher bleaching severity)? Again, see comments above and below to clarify the specific calculation used for this metric.

Thank you for your insightful comment about the Δ DHW, you are right, we already modified the figure. See below.

Methods:

Line 287 change to "sensitivity of corals to bleach[ing]"

Lines 345 and 346: Five different metrics were calculated to describe the sensitivity of corals to bleach, based on species composition and reef diversity (Table 1).

Lines 316-317 see comments above concerning the need for a more detailed explanation of the ROTC_{clim} variable, specifically whether the three month period was the same for each year or calculated for each year individually based on the average weekly maximum temperature as in Chollett et al. 2014

We apologize for any confusion arising from the discrepancy in our initial definition and have now amended it to align with the accurate methodology described.

We included some of the modifications regarding this metric:

Table 1: "The Rate of Temperature Change (ROTC) reflects the trend in temperature change over 84 weeks during summer⁹. ROTC_{clim} is the average annual ROTC values for the period 1985-2012 (termed rate of seasonal warming in Chollett et al. 2014)⁹."

Lines 389 to 393: "The rate of temperature change (ROTC; termed rate of seasonal warming in Chollett et al. 2014)⁹ was calculated for each site, and we (1) calculated the weekly average temperature; (2) identified the maximum; (3) calculated the rate of temperature for the year as the trend for the previous 3 months; (4) in the case of ROTC_{clim} we calculated an average of all rates for that site from 1985-2012."

Table 1 DeltaDHW field. Suggest additional clarification here as well. Particularly, which

value was subtracted from which. In the table, it sounds as if the formula was this: $\text{maxDHW}_{\text{lastyear}} - \text{maxDHW}_{\text{currentevent}}$ but in the text it sounds like it's the opposite $\text{maxDHW}_{\text{currentevent}} - \text{maxDHW}_{\text{lastyear}}$. It makes a big difference to the interpretation. Also the Hughes et al 2019 reference does not use the delta DHW terminology so is misleading to cite as the basis for this indicator using that name, perhaps a "sensu" or "building on" preface to the Hughes citation is appropriate to add.

Thank you for your insightful comment on the Δ DHW equation. We've reviewed Table 1 and the text, acknowledging potential confusion. Δ DHW represents the difference in DHW between the current bleaching event and the previous year, as outlined by Hughes et al. (2019).

Table 1

Difference between the maximum observed value of DHW in the current event up to the sampling date and the maximum observed value of DHW in the last year (building on Hughes 2019¹⁹).

Lines 416 to 420: The second metric was the ' Δ DHW', this metric represents the difference between the maximum observed value of DHW in the current event up to the sampling date and the maximum observed value of DHW in the last year (building on Hughes 2019¹⁹), this metric provides information on the relative magnitude of the heat stress event as a function of the previous year's exposure.

Reviewer #2 (Remarks to the Author):

This is beautiful paper – really well illustrated and discussed, and covering a complex field concisely. The results are very important in the literature, showing a well-constructed analysis of multiple drivers/factors affecting coral bleaching. It provides a very important next-step in this field of research.

There are some problems that I address in more detailed comments – these do need to be addressed for the paper to not fall into some of the traps that are common in this field (particularly, having the 'final say' on what controls coral bleaching and that results must be relevant everywhere).

I certainly recommend for publication with these points addressed.

Thank you for your positive feedback and recognition of the importance of our paper. We apologize for the delay in the response process. We appreciate your thorough review and acknowledgment of the complexities within the field of coral bleaching research. We will carefully address the specific points you've raised to ensure the robustness of our conclusions and to avoid common pitfalls in the field.

There are no line numbers, so I copy out text fragments to help locate my comments.

Abstract

The paper presents good discussions on the conflicting results that are in the literature (e.g. deep reefs being a refuge vs. being more vulnerable) ... it all depends, but falls a bit into the same trap proposing their results can be "transferable to other regions" ... the statement in the abstract is good, "tools to be explored by coral research and conservation programs", but in some places in the text falls into the trap of claiming to provide a predictive algorithm that can be applied elsewhere. See more detailed comments.

“Here we demonstrate a regional warning system” ... no, this does not demonstrate a warning system. It analyses factors associated with variation that can be used in a warning system. The study looks at factors associated in an event(s) that happened – how well this will perform in a predictive/warning system needs to be developed and tested. So the orientation of the primary claims of the paper need to be shifted to reflect this. The last sentence in the intro is correct that there is potential for this use, not demonstration of it here “.. with a high potential for use as an early warning system for the Mesoamerican Reefs transferable to other regions”. And see comment below, what is ‘regional’ – the spatial scope of the study must be stated in the abstract.

"multiple drivers of coral bleaching" – some of these are ‘drivers’, but some are factors affecting the bleaching response. Need to be more explicit about these roles.

a

The last sentence in the abstract suggests using this in other regions, which is great, but the abstract (nor the paper) is not explicit about what a ‘region’ is – the Caribbean, the MAR? The abstract must state the scale of this analysis, ie. the MAR.

We sincerely appreciate your valuable insights and thoughtful feedback on our work. Your comments, along with those from other reviewers, have been instrumental in refining our abstract. As a result of this constructive input, we have revised the abstract, and the updated version is as follows:

Lines 17 to 27: “Coral bleaching, a consequence of stressed symbiotic relationships between corals and algae, has escalated due to intensified heat stress events driven by climate change. Despite global efforts, current early warning systems lack local precision. Our study, spanning 2015-2017 in the Mesoamerican Reef, revealed prevalent intermediate bleaching, peaking in 2017. By scrutinizing 23 stress exposure and sensitivity metrics, we accurately predicted 75% of bleaching severity variation. Notably, distinct thermal patterns—particularly the rate of temperature change and various heat stress metrics—emerged as better predictors compared to conventional indices (such as Degree Heating Weeks). Surprisingly, deeper reefs with diverse coral communities showed heightened vulnerability. This study presents a framework for coral reef bleaching vulnerability assessment, leveraging accessible data (including historical and real-time sea surface temperature, habitat variables, and species composition). Its operational potential lies in seamless integration with existing monitoring systems, offering crucial insights for conservation and management.”

Main text

“We analysed remote sensing data and 266 *in situ* observations recorded during the seasonal “bleaching window” ... see later comment on ‘samples’ vs. observations. The observation that is analysed is each coral colony record, not each reef sample ... If I’ve got this wrong, perhaps it’s the % bleaching levels and BSI that are analysed at reef level, then this needs to be made clearer.

See comment below.

Thank you for your comments we clarified this in two sentences, see below.

Lines 69 to 71: “To evaluate this hypothesis, we analyzed remote sensing data and 266 *in situ* reef level samples/observations recorded during the seasonal ‘bleaching window’, August to December (Supplementary Fig. 1) along the Mesoamerican Reef (MAR)⁴², during 2015-2017.”

Lines 95 to 97: “We assessed 55,177 colonies using the following categories: pale (with significant discoloration), partially bleached (bleached tissue present), and fully bleached (with > 90% bleached tissue). Over half of the 266 reef-level samples/observations recorded from 2015 to 2017 reached bleaching severity values ≥ 28.0 (i.e., reefs with at least 28% affected colonies; Fig. 1a).”

Species with thin tissues and flat-extraplanate colony morphologies – is “extraplanate” a correct word, or should it be ‘explanate’

Thanks for your suggestion, we acknowledge that the correct word is explanate. However, we modified the sentence and used “plated colony morphologies” instead.

Lines 249 to 252: “Species with thin tissues and plated colony morphologies such as those of the genus *Agaricia*, were, as expected, among the most sensitive, while the branching *Acropora palmata* showed the lowest bleaching severity within the most abundant species (Fig. 3).”

“Our findings do not question this conclusion but highlight the need for a better understanding of the cellular mechanisms behind the physiological disturbance of this symbiotic association and all the biological processes that can confer stress resistance at the symbiont, host, and holobiont/microbiome levels.” – or is it just that there are a great many factors influencing these processes and its more important to base analysis on observations rather than theory/extrapolation from elsewhere and different times/events?

Completely agree with your suggestion, and we modified the text including your comment.

Lines 262 to 267: “Our findings also highlighted the need for a better understanding of the cellular and functional mechanisms behind the physiological disturbance of this symbiotic association and all the biological processes that can confer stress resistance at the symbiont, host, and holobiont/microbiome levels. Multiple factors influence these processes, highlighting the importance of basing analyses on observations and not generalizing theoretical or experimental results applied to different scales and regions. The exact combination of stressors may vary with each bleaching event in space and time.”

“Thus, global bleaching predictive models can be refined on regional scales when regional data such as these are available” – again, be explicit about what is meant by “regional” – the Caribbean, part of the Caribbean, the MAR? The implications for developing a warning system and its spatial reach are very different for these.

Thanks for your comment, we completely agree. In this sense, we modified the text, and we clarified that the model was created for operational use in the MAR but this could be transferable to predict the vulnerability of corals in the Caribbean coral reefs.

Lines 301 to 315: “In conclusion, our study emphasizes the importance of understanding the key drivers of coral bleaching by integrating various reef sensitivity and heat stress metrics. We have identified multiple novel thermal patterns that better predict coral bleaching on MAR reefs and propose a transferable model to predict the vulnerability of coral reefs to bleaching in the Caribbean coral reefs. This model can facilitate the development of emergency responses and conservation strategies through an automated warning system (Fig. 4). The model not only has theoretical application, because reefs that have information on diversity, depth, and historical and real-time SST data an operational prediction can be made. This will allow the creation of a high-precision early warning system with adaptive learning capacity, which may be of great relevance to coral reef managers. While there are already operational warning systems for coral bleaching, validation with in situ observations is needed to improve their accuracy. Long-term coral bleaching data collected in different reef regions can be used to refine global bleaching predictive models on ecoregional

scales. We highlight the need for collaborative coral bleaching monitoring networks along with emergency response plans and data-sharing platforms (such as <https://www.healthyreefs.org/cms/> and <https://www.agrra.org/>). Given the increasing frequency and severity of coral bleaching events, improving these early warning systems for managers and conservation planning efforts is crucial.”

Methods

“Considering the three sampling periods, 266 observations were obtained at the reef level, for 69 sites in 2015; 104 sites in 2016; and 93 sites in 2017” – this is confusing – an ‘observation’ is more strictly each colony observed/documentated. Is this better to state as ‘samples’, or reorganize as “266 reef level samples/observations were obtained, 69 in 2015”

Thanks for the suggestion we modified the text as suggested.

Lines 333 to 335: “Considering the three sampling periods, 266 reef-level samples/observations were obtained, 69 in 2015; 104 in 2016; and 93 in 2017.”

These statements are only understandable to people who know the details of these, consider explaining for a broader audience: “N1 of Hill numbers” and “The effective species is the number of species in the community in which all species were equally common, this represents the true diversity without considering the less abundant or “rare” species”

We recognize the lack of clarity in the sentence related to the diversity index metric, in this sense, we modified some parts of the manuscript to better explain the use and the meaning of the N1 of Hill’s numbers as the diversity metric used. See below.

Table 1: The diversity of corals calculated from Hill's number one is equal to Shannon’s diversity exponent, this represents true diversity without considering the less abundant or “rare” species⁵⁴.

Figure 2 legend in lines 154 to 157: “Diversity is the Hill's number one expressed in effective species. Under this framework of Hill’s numbers, the diversity of a community is measured as the effective number of species in it, which can be understood as the number of species in a virtual, perfectly balanced community, in which all species are equally common, and in which the average relative abundance of the species in the real community is conserved⁵⁴.”

Fig 4 – the integrated diagram is good, but the proximity of the ‘+’ signs within the yellow box and the one outside is confusing. Perhaps move the inner ones to the left, mirroring the blue box.

Thanks for the suggestion we completely agree and we modified the text as suggested. See below.

Supplementary Figure 3. Caption – should the second sentence start with “We”?

Thanks for the suggestion we agreed and we modified it as suggested.

Supplementary Figure 3. Detrended correspondence analysis (DCA) ordination plot of the coral community. Gray points represent the site's ordination based on coral species composition. Only the 28 most abundant species are shown, with an image of the typical morphology of some coral species.

Reviewer #3 (Remarks to the Author):

The paper is a description of the bleaching surveys undertaken in the MesoAmerican Reefs from 2015 to 2017. The authors describe the bleaching intensity of the taxa in space and time and evaluate a number of thermal and depth metrics in predicting the response levels. This represents a large and well coordinated program of study that is unique in the Caribbean. The original part of this study is the use of the machine learning algorithm to evaluate the potential influence of various thermal metrics. The produces a good fit to the data, which is a good outcome and considerably improves on previous efforts (not cited) to do this on a large scale. Below I list a number of the weaknesses of the paper that may be rectified with a major revision.

We are very grateful for the reviewer's comments and have addressed the suggested weaknesses in detail.

The paper is descriptive and not really driven by any specific or alternative or competing stress theories or hypothesis. It has a data-mining approach that leads to a weak organization and descriptive conclusions.

We respectfully disagree with this comment. We do not think it is descriptive and although it lacked clarity in describing and establishing the hypotheses and theories, many of these are included and described in the introduction. The fact that we are using Machine Learning methods does not directly imply that it is a data mining approach, all the variables chosen for the analysis and selected in the model include the ecological explanations, and physiological and biological processes that we were able to document and/or discuss.

Additionally, the organization is weak in terms of focusing on the original findings and the main points that are contributing to science, as opposed to a description of what occurred in MBR reefs during this period. The authors need to rethink what should be in the main and supplementary text and elements.

We appreciate your comments and have improved the organization and writing of the whole manuscript.

Many of the variables are what are typically evaluated by NOAA in terms of their metrics of stress. But, the literature has moved on from these early recommendations of NOAA and is examining more variables and testing more theories of stress. The paper therefore feels dated as the citation is not current and there is little or no connection to a number of important recent studies of bleaching and coral mortality. The authors need to search the literature since 2016 and incorporate these newer papers into their approach. Many recent studies are finding variables that are important that are not examined here. Many of the studied variables here are not well described and confusing and only seldom are equations given.

We appreciate the reviewer's comment and try to improve and clarify many of the aspects discussed by the reviewer.

However, we emphasize that the work focuses primarily on thermal and site-based variables (including species composition, depth, and diversity), as these are the most readily available variables for operational early warning systems. In this sense, this work did not intend to include variables other than these as is explained in the manuscript.

On the other hand, we disagree that the same typical variables are used in this work. This paper includes many more recent thermal metrics outside the NOAA operational framework. In the same way, we clarify that both at the time of submission of the manuscript and in this version many of the papers cited are from after 2016. In fact, 41 of the 96 papers cited are from after 2016, which represents about 45% of the references.

Table 1 is a good place to organize more effectively. This makes it hard to confirm the methods and interpret the data, which leads to weak support for the paper as science. What effect might the many issues of variables studied here and elsewhere have a results, interpretation, and comparison with the larger literature?

We are very grateful for this comment, we fully agree and have brought the table to the front in the results, added suggested information, and modified most of the paragraphs mentioned.

I have limited my detailed comments as I think these larger focus, context, and compositional issues will need to be addressed before a decision can be made.

Abstract

L20 – not more frequent but longer duration – see Skirving et al. 2019

Thank you for your comment. Our abstract has been modified to read the following. We have removed this statement.

Coral bleaching, a consequence of stressed symbiotic relationships between corals and algae, has escalated due to intensified heat stress events driven by climate change. Despite global efforts, current early warning systems lack local precision. Our study, spanning 2015-2017 in the Mesoamerican Reef, revealed prevalent intermediate bleaching, peaking in 2017. By scrutinizing 23 stress exposure and sensitivity metrics, we accurately predicted 75% of bleaching severity variation. Notably, distinct thermal patterns—particularly the rate of temperature change and various heat stress metrics—emerged as better predictors compared to conventional indices (such as Degree Heating Weeks). Surprisingly, deeper reefs with diverse coral communities showed heightened vulnerability. This study presents a framework for coral reef bleaching vulnerability assessment, leveraging accessible data (including historical and real-time sea surface temperature, habitat variables, and species composition). Its operational potential lies in seamless integration with existing monitoring systems, offering crucial insights for conservation and management.

L26 – not sure I understand if these are in the year of the bleaching or historically. What is heat stress here, needs a definition? Studies like this often distinguish between historical and current metrics. See Sully et al. 2020.

We used both climatological and historical values as well as current values. We appreciate the comment and specify it better in the text and Table 1.

Lines 20 to 23: Seven metrics of stress exposure and sensitivity predicted approximately 75% of the variation in bleaching severity. We identified specific thermal patterns (including historical and current metrics) that are better predictors of coral bleaching events than the commonly used heat stress metrics.

One gets the feeling these reefs were already exposed to thermal stress and therefore deep reefs are now being affected. Maybe something about the contextual location and time period of this study would be useful for context. The abstract is somewhat mysterious because it is so short.

We appreciate the comment, but we do not consider that this is something so fundamental to include in the abstract, besides we have reached our limit in the abstract. On the other hand, we give some context by explaining that we analyzed the period from 2015 to 2017 and we did it in the Mesoamerican Reef System within the wider Caribbean.

I also wonder if one could actually get these data prior to a bleaching event. It seems this is post bleaching analysis with data that may not have been available prior to the bleaching and therefore theoretical and not applicable to actually predicting bleaching. Can this be discussed later in the text?

We are grateful for this comment, which we consider fundamental and very useful for the applied conservation of the region. We emphasize that our results and method can indeed predict bleaching almost in real-time, although not in advance. Once we have the most recent SST information, as

well as historical information of the site, it is possible to make the prediction. It does not necessarily have to happen and then predict.

It is in this sense that we have included more details in different parts of the manuscript, among which are the sections included below.

Lines 24 to 27: “This study presents a framework for coral reef bleaching vulnerability assessment, leveraging accessible data (including historical and real-time sea surface temperature, habitat variables, and species composition). Its operational potential lies in seamless integration with existing monitoring systems, offering crucial insights for conservation and management.”

Lines 80 to 87: “Based on these results, we propose a model to predict coral reef vulnerability to bleaching, with a high potential for use as an early warning system for the Mesoamerican Reefs transferable to other ecoregions in the wider Caribbean. It can also be used to predict reefs with intrinsically more resistance to bleaching for conservation planning purposes. The operational use of our model is facilitated by the accessibility of essential data sources, including historical and actual sea surface temperature data, habitat variables, and species composition information. These readily available datasets enable efficient implementation and integration into existing monitoring systems, providing valuable insights for coral reef conservation and management efforts.”

Lines 302 to 309: “We have identified multiple novel thermal patterns that better predict coral bleaching on MAR reefs and propose a transferable model to predict the vulnerability of coral reefs to bleaching in the Caribbean coral reefs. This model can facilitate the development of emergency responses and conservation strategies through an automated warning system (Fig. 4). The model not only has theoretical application, because reefs that have information on diversity, depth, and historical and real-time SST data an operational prediction can be made. This will allow the creation of a high-precision early warning system with adaptive learning capacity, which may be of great relevance to coral reef managers.”

Introduction

A number of recent studies are suggesting geographic position or some regional differences in bleaching. Wondering why this is not mentioned as it can be a dominant factor? Perhaps this is not so important for MBE but was found for some ecoregions where coral cover and bleaching were evaluated (Vercammen et al. 2019; McClanahan et al. 2020).

Thank you, we have updated the following.

Lines 54 to 57: Even the geographic position or region in which reefs are located may be a good predictor of coral bleaching, highlighting the potential spatial variation in stress exposure or adaptation processes in corals across a geographical gradient^{15,16,20,22}.

L73- given there are no methods until here in the reading, can a bit more be said about what the BSI is? Otherwise, the rest of the text is hard to interpret.

Lines 91 to 95: The bleaching severity index used here incorporates coral response categories that have previously been used in the MAR and globally^{15,23,46,47}. The bleaching severity is calculated from the sum of the proportion of colonies in each response category, weighting each category according to its ecological impact, with the proportion of fully bleached colonies having a higher weight and the proportion of pale colonies having a lower weight.

L76- what is an observation? I assume it is some sample of many colonies but this seems important to make clear early.

Lines 97 to 99: We assessed 55,177 colonies using the following categories: pale (with significant discoloration), partially bleached (bleached tissue present), and fully bleached (with > 90% bleached tissue). Over half of the 266 reef-level samples/observations recorded from 2015 to 2017 reached bleaching severity values ≥ 28.0 (i.e., reefs with at least 28% affected colonies; Fig. 1a). The most common categories were partially bleached and pale (Fig. 1a; Supplementary Table 1).

L84 – not sure what spatial footprint means?

We remove these words and describe the spatial patterns observed instead.

Figure 1 has so many panels that it is quite confusing to interpret. The problem with Nature paper is putting multiple figures on a single figure. Is A) summarized for all years? B) – what is index? Not really sure what C) is even after reading the methods. Seems like a supplementary figure. How important is all of these panels for the main conclusions of the paper? Maybe less information here and more in the supplement would work better. If the main point of this figure is that 2017 was the worst year, this is too much information. Could a simpler graph that shows the severity in space and time in 2 panels all that is needed to make the main point? Then most of these descriptive details can go to the supplement.

Line 112-118 Thank you for this important comment, We have removed various graphs and have left only the below:

Fig. 1. Spatiotemporal variation of coral bleaching in the Mesoamerican Region during the years 2015-2017. (a) Distribution of the Bleaching Severity Index (BSI) values and the proportion of colonies in each bleaching category, each year, for all the reefs sampled. Information on the BSI's statistical descriptors and the categories defined for coral bleaching can be found in Supplementary Table 1. (b) Maps illustrate the spatial distribution of the severity of coral bleaching in each year, for all sampled reefs.

Does figure 2 really require so many abbreviations? It just makes it hard to follow if a reader is not familiar with these abbreviations. At least bring the table forward so readers can see them, their definition, and maybe equations.

Thank you for this comment. We have moved Table 1 to Line 139 to 141, before figure 2, so the description of the abbreviations is included prior, to help guide their definition.

The negative relationship with ROTC is provocative finding given all models to date use it to predict disasters for reefs, i.e. Hoegh Guldberg 1999, etc. This seems important to spend some text on, as the future models are likely to be quite poor predictors if this is more widespread.

Thank you for this comment. We updated this paragraph in Lines 158 to 185.

We found the climatological rate of temperature change from 1985 to 2012 ($ROTC_{\text{clim}}$) to be the main driver of the severity of coral bleaching (Fig. 2a). $ROTC_{\text{clim}}$ expresses the mean trend of temperature change over the spring-to-summer transition (i.e., three months before the maximum weekly average temperature) along the period 1985-2012, this metric has also been termed the rate of seasonal warming⁹. The reefs with the highest $ROTC_{\text{clim}}$ had the lowest bleaching severity (Fig. 2d). We found that higher climatological rates of temperature change in summer could be a key contributor to increasing coral resistance to bleaching. Our findings suggest that corals present in reefs with a thermal history characterized by higher and more rapid seasonal change were more resistant during the bleaching event. Environmental seasonality is particularly important for corals, as seasonal variation in irradiance and temperature directly impacts the rate of photosynthetic carbon fixation. In contrast, temperature changes also induce important adjustments in the heterotrophic metabolism. Corals acclimated to areas with moderate to high thermal variations are accustomed to physiological variations and adaptations which may assist their ability to acclimate to thermal stress-associated bleaching events^{16,17,22,30,57-59}. Several Caribbean corals have demonstrated an ability to express two different coral holobiont phenotypes, winter, and summer, with contrasting susceptibility to bleaching under similar heat stress exposure³⁰. The rate of temperature change in spring to early summer may determine when the summer coral phenotype is fully achieved, and therefore when the coral is prepared to cope with heat stress³⁰. This implies that a more robust response could be expected when accumulated heat stress coincides with the complete expression of the summer phenotype³⁰. However, when the heat stress event occurs before the complete expression of the summer coral phenotype, or if moderate or acute heat stress is prolonged for too long, different physiological responses can be expected³⁰. This could be a physiological mechanism that could explain the results of our work. Still, more research is needed to explain the contribution of $ROTC_{\text{clim}}$ to coral bleaching during heat stress events. This is of

relevance as it could represent an emerging pattern of potential resilient populations, which may modify the perception we have regarding previous existing models^{9,17,60}, especially if this is an emerging pattern in all reefs worldwide. In our results, the ROTC_{clim} represents a climatological thermal variability metric without collinearity with most of the indicators of heat stress. It is important to mention that high rates of temperature change do not necessarily result in high exposure to heat stress, and sites with high rates of temperature change but low heat stress could be considered as potential refugia^{9,16,57,61}.

P125 – this discussion of summer and winter phenotypes seems to assume this is true for all corals. How well studies and broad is this seasonal pattern concept? This seems speculative and not really critical for the paper.

Thank you for the suggestions and commentaries. We also described the following text description.

Seasonal fluctuations in light and temperature significantly impact organism physiology, including symbiotic coral regions (Scheufen et al., 2017). While studies on this seasonal variability are limited, other coral traits such as pigmentation and symbiont numbers have been extensively researched (Brown et al., 1999; Fitt et al., 2000; Fagoonee et al., 1999; Warner et al., 2002). The distinction between seasonal phenotypes characterized by Scheufen et al. (2017) reveals different responses to temperature and heat stress, highlighting the importance of considering annual variability in coral physiology when studying bleaching susceptibility and severity. These authors documented significant physiological adjustments in all coral metabolic rates (photosynthesis, calcification, and respiration) as well as in the four coral species analyzed. At higher latitudes, such for example the Florida Keys, the same coral species will experience larger seasonal adjustments in their metabolism, for similar depths, relative to the variation reported by Scheufen et al (2017). Thus, the answer to the reviewer's question is YES, only in sites with very stable environmental conditions and where this climatic driving force is absent, these physiological adjustments can be ignored. This conclusion is not a theoretical-speculative interpretation, but a well-known fact derived from the understanding of how temperature and light affect the physiology of symbiotic corals.

Concerning 'how many studies have investigated this seasonal variability', it is unfortunate that the physiological variability of corals has received so little attention so far, but this is not the case for other coral traits closely related to coral physiology, such as: i) coral pigmentation (Brown et al. 1999; Fitt et al. 2000); ii) the number of symbionts (Fagoonee et al. 1999; Fitt et al. 2000); photochemical efficiency of photosystem II, Fv/Fm (Warner et al. 2002); and coral net calcification (i.e., Barnes and Lough 1989; Carricart-Ganivet et al. 2000; Falter et al. 2012). Indeed, seasonal changes in coral calcification are the cornerstone of coral sclerochronology. The relevance of the distinction between the two different seasonal phenotypes characterized by Scheufen et al. (2017), is that they present different responses to temperature (figure 4 in Scheufen et al. 2017), as well as to similar levels of heat-stress along 10 days of exposure. We are aware that the ecological implications of the regional (spacial) variation in temperature are better understood and accepted in coral ecology, than this annual variability, but why changes in temperature of similar magnitude cannot be considered as important? The physiological implications of this temporal variability for coral performance are equally relevant, as they involve similar metabolic adjustments in coral physiology.

Lines 154 to 181: We found the climatological rate of temperature change from 1985 to 2012 (ROTC_{clim}) to be the main driver of the severity of coral bleaching (Fig. 2a). ROTC_{clim} expresses the

mean trend of temperature change over the spring-to-summer transition (i.e., three months before the maximum weekly average temperature) along the period 1985-2012, this met

L187 – why not use historical data as a predictor? I think this was done in the Sully et al. 2019 Nat. Comm. paper.

Thank you for this comment. The main reason we used this dataset as a predictor, is based on the groundwork and data availability. We reviewed data sets available for the region and found there were many gaps in the information of different species such as Swain et al 2016. Moreover, we are certain that this data set represents the largest effort at this scale in the MAR regions.

One gets the feeling that some taxa, like *Acropora*, do not bleach but die. This is one option a maybe this bleach/no bleach vs die/no die needs some consideration. Can you use the mortality data to evaluate this for the studied taxa? Perhaps the BSI is not good at distinguishing bleaching versus dying?

Thank you for the comment, we agree with the reviewer, that the difference between corals that bleach or not versus those that undergo mortality deserves special consideration and discussion. This applies to all taxa and even to results at the scale of spatial variation. Unfortunately, we do not have mortality information associated with the bleaching data, across all the sites in the 4 countries of the MAR. Therefore, we cannot make this assumption based on our data.

The discussion section would benefit from stronger English composition. It rambles and is not highly organized around key conclusions, caveats, and context.

Methods

L272-275 – this site selection text is not very clear as per the decisions, only that they had been done in the past. Please explain how or why they were originally selected. Is there any experimental design in site selection?

In Lines 321 to 334 we better describe the Field methods and study area to include the following: “To assess bleaching severity, the “bar-drop” method was employed to survey a minimum of 150 to 200 individual coral colonies using a 1m PVC bar with 5 marks every 25 cm. The bar was haphazardly placed across the reef after 3 - 4 kick cycles⁴⁷. Corals were identified at the species level and assessed using three bleaching categories: pale colony, partially bleached colony, and whole colony bleached with over 90% of live tissue affected; and one category for non-affected colonies, which were recorded as ‘normal’. Sampling was conducted throughout the whole MAR region in three periods: October-November 2015, 2016, and 2017. Sites were selected based on information from previous monitoring programs in the region. Site selection was stratified according to cross-shelf position (e.g., bank reefs, patch reefs, fringing reefs), the reef zone (e.g., crest, forereef), depth, and wind exposure (e.g., wave exposure). Most of the selected sites had consistent information in other regional databases (e.g., Atlantic Gulf and Rapid Reef Assessment-Healthy Reefs Initiative, protected areas) and we prioritized areas based on the experience and feasibility of the surveys achieved by local experts. The monitoring was conducted by trained volunteers from various partner institutions of the Healthy Reef Initiative within Mexico, Belize, Guatemala, and Honduras. Considering the three sampling periods, 266 reef-level samples/observations were obtained, 69 in 2015; 104 in 2016; and 93 in 2017.”

L292- I believe this RFI values needs an equation to be better understood.

A revised description of the RFI value has been included in Lines 353 to 360: As an approximation to characterize structural complexity, the Reef Functional Index (RFI; Equation 3) was calculated based on the summation of the abundance (Number of coral colonies; N_{cci}) multiplied by a functional coefficient (F_{ci}) of each species for the reef site⁷⁶. The functional coefficient considers multiple morphological and growth characteristics of each coral species present in the region⁷⁶. When species did not have a functional coefficient, the value available for congeners was used (e.g., *Solenastrea hyades* was used for *Solenastrea bournoni*). Colonies were not considered when a value for the species could not be assigned (i.e., *Oculina diffusa*).

$$RFI = \sum \left(\left(\frac{N_{cci}}{100} \right) * F_{ci} \right) \quad (3.)$$

P309 – I find most of these metrics not well explained and wonder if equations would help. For example, is this a new type of MMM? It is not that clear as described and therefore confusing as previous MMM are the 3 hottest months of the year over time. How can one take an average of a hottest month over time? This would be only 1 month. I think there is some poor writing going on here but this text has to be clear to be evaluated.

This section has been updated to strengthen the writing, particularly in Lines 378 to 420: To characterize the effect of different descriptors of heat stress on the response of corals and reefs and the expression of bleaching, 17 metrics were calculated based on the variation of Sea Surface Temperature (SST; Table 1). SST and heat stress metrics were obtained from the CoralTemp database of Coral Reef Watch (<https://coralreefwatch.noaa.gov/product/5km/index.php>). This database has a resolution of ~ 5 km and a period from 1985 to the present, with a daily frequency⁸⁹. Additionally, Maximum Monthly Mean (MMM) values were obtained from the same database.

Moreover in Lines 396 to 401. Furthermore, we calculated novel metrics to characterize the accumulated, acute, and chronic heat stress (Table 1). The indicators generated are based on two main metrics known as Hotspot (HS; Equation 5) and Degree Heating Weeks (DHW; Equation 6), which consider thermal anomalies or heat accumulated above the MMM over a certain period³⁸. Hotspots (HS) represent daily positive anomalies above the MMM³⁸. Heritage DHW quantifies heat stress by summing up positive daily anomalies over 1 °C above the MMM over 84 days (12 weeks), divided by 7 to express values per week³⁸.

I like table 1 and wonder if it could be in the main text with a column for the equations and maybe citations of there they were first used? What the ranges of values are and what are the units, etc. This would help greatly with interpretation. See Nature Climate Change 9:845-851 for a previous example published in the Nature series.

Thank you very much for your comments, we have followed your suggestion and have incorporated this table in the main text. We have also included the range of values, and the units, as well as some small notes. However, we do not have the space to include the equations, these were incorporated in the methods in the cases where we felt it was necessary to include them.

Table 1 is many heat stress metrics but not much consideration of the other modifying variables including light, water quality, etc. Is there some reason for this focus? It seems there are not really any hypotheses driving this paper but rather a bit of a data mining procedure. Can the data mining aspect be reduced?

The main reason for the current approach is its operational intent, with the main aim of using standardized data sets that are readily available and easy to connect with other data platforms across the MAR region. In the case of the MAR, thermal variables from SST data derived from satellite

sensors, habitat variables, as well as species composition information, are all currently available. We also included this focus in the introduction of the Manuscript.

P325 – best to start a paragraph saying why you are doing this text between years? It seems tacked in otherwise.

This was updated in

Lines 422 to 431: To identify temporal differences in bleaching patterns between the three years under consideration, a Yuen's test (robust t-test) on trimmed means for dependent samples was carried out⁹⁰. The temporal comparison was performed considering only the sites re-sampled in both years in the paired comparisons. Yuen's test was chosen because the values of bleaching severity and bleaching categories in re-sampled sites generally were not homogeneous in variance or normality. The test of normality applied was Shapiro-Wilk. Levene's test was also used to check the homogeneity of the variances. For Yuen's test, a trimmed value of 0.10 was used, eliminating 10% of the outliers on each side of the distribution for a more robust comparison. This test was performed with the "yuend" function available in the "WRS2" library⁹⁰ of the R statistics program⁸⁸. The function used estimates of an explanatory measure of effect size, this was realized using a robust heteroscedastic approach for two or more groups⁹⁰.

GBM sounds a lot like a BRT. Are they the same?

Indeed, GBM (Gradient Boosted Models) and BRT (Boosted Regression Trees) are essentially synonymous, representing variations of the same algorithm. While GBM encompasses a broader range of boosting techniques, BRT specifically refers to boosting applied within the realm of regression trees. We include this clarification to provide a clearer understanding in Lines 77 to 79: "Gradient Boosted Models (GBM, also known as Boosted Regression Trees), a machine learning algorithm, was utilized to identify relative direct associations, non-linear relationships, and interactions^{15,43–45}."

L350 – Best to give short results of what the final variables selected were and why. Why not put results in table 1?

We have already included in Table 1 the differentiation between selected and non-selected variables. However, further explanations were omitted due to limitations in space and the complexity of the variables' interpretations, which may not be effectively conveyed through tabular presentation.

There are many relevant recent papers that are not cited, even ones directly related to the Caribbean let alone the larger reef locations.

Welle, P. D., Small, M. J., Doney, S. C., & Azevedo, I. L. (2017). Estimating the effect of multiple environmental stressors on coral bleaching and mortality. *PLoS One*, 12(5), e0175018.

Thank you for the great recommendation, it has already been included in different parts of the manuscript. However, it is also necessary to say that the article is not a literature review and we already have many references included.

Point-by-point response to comments on "Underlying drivers of coral reef vulnerability to bleaching in the Mesoamerican Reef"

October 2024

We would like to thank the Reviewer and the Editor for their help and special attention during the review. This review allowed us to improve the grammar of some specific sections of the manuscript, and to clarify the robustness and relevance of the analysis.

We provide a point-by-point response to the comments made in the manuscript. Reviewer's comments are in black; authors' responses are in dark blue text. The main additions/editions to the document have been included in this archive.

REVIEWERS' COMMENTS:

Reviewer #4 (Remarks to the Author):

To the editor and authors:

I have read the revised manuscript and the rebuttal document. In my opinion, the authors have responded thoughtfully and fully to all the major suggestions made by the three reviewers. I do not have any concerns with their response, I see no conceptual flaws, the conclusions are original, and the paper contributes new knowledge that will appeal to a broad science-to-management audience involved in coral reef issues. Some of the prose, especially in the results / discussion section, remains hard to read and is repetitive in places. To remedy some of this, I here make a few suggestions:

Thank you for your detailed and constructive feedback. We appreciate your positive assessment of our revisions and the thoughtful suggestions provided to improve the manuscript. In response, we have revised the prose for clarity and conciseness in the results and discussion sections, addressing the issues of repetition and ambiguity.

Line 158: The "rate of temperature change from 1985 to 2012" is a misleading way of stating what this variable is when it is defined differently in the next sentence. Recommend change to something like: "We found the climatological seasonal-warming rate (*sensu* Chollet et al. 2014) from 1985 to 2012 (ROTC_{clim}) to be the main driver of the severity of coral bleaching (Fig. 2a). ROTC_{clim} expresses the mean trend of temperature change over the spring-to-summer transition (i.e., during the three months before the maximum weekly average temperature) from 1985-2012. (and delete "this metric has also been termed the rate of seasonal warming9).

Thank you for pointing this out. We have revised the sentence to:

Lines 135 to 138

"We found the climatological seasonal-warming rate (*sensu* Chollet et al. 2014⁹) from 1985 to 2012 (ROTC_{clim}) to be the main driver of the severity of coral bleaching (Fig. 2a). ROTC_{clim} expresses the mean trend of temperature change over the spring-to-summer transition (i.e., during the three months before the maximum weekly average temperature) from 1985-2012."

Line 162, delete the repetitive sentence “We found that higher...”

We have deleted the repetitive sentence. Check lines 135 to 40.

Line 197, replace “in this sense” with “; thus, it is...”

We have replaced “in this sense” with “; thus, it is...” as per your suggestion (line 172).

Line 201, delete the repetitive sentence “These differences can be addressed...” You just stated that above.

We have deleted the repetitive sentence. Check lines 173 to 179.

Line 214, “the difference between two consecutive years” is ambiguous, change to “the difference between most recent and previous year.” Then deleted repetitive sentence in line 216: “These were sites with...”

We have clarified “the difference between two consecutive years” to “the difference between the most recent and previous year” and deleted the repetitive sentence. Lines 188 to 198.

Lines 216 to 225, this section on delta DHW is still confusingly written and needs refinement. State more clearly and simply what Hughes et al (2019) showed. Delete sentence in line 225 “This result is probably...”

We have refined this section on delta DHW for better clarity, clearly stating what Hughes et al. (2019) showed. We also deleted the sentence in line 225.

Lines 190 to 198:

“High values of Δ DHW indicate much greater heat exposure during the recent event compared to the previous year, which corresponded to a considerable increase in bleaching severity, particularly at values above 6 °C-weeks. This aligns with the findings of Hughes et al. (2019)¹⁹, who demonstrated that the severity of bleaching in 2017 on the Great Barrier Reef was significantly influenced by the geographic patterns of bleaching observed in 2016. The study showed that reefs with high heat exposure in 2016 exhibited reduced bleaching in 2017, even under similar thermal stress, suggesting an ecological memory effect. This response may result from acclimatization, adaptation, or changes in species composition, highlighting the potential physiological or ecological memory retained by corals due to past heat stress^{19,20}.”

Line 244, perhaps the circular nature of this conclusion should be acknowledged. If I understand correctly, the bleaching response data were used to define SI for each species, and then related back to observed bleaching severity, is this correct? Or were different years’ data used to define SI?

We have acknowledged the circular nature of the conclusion and clarified whether the same or different years’ data were used to define SI and relate it back to bleaching severity.

Lines 213 to 216:

“While this index was derived from data on bleaching responses during these years for the whole MAR, it was also used to assess how the composition of species with different sensitivities could predict overall reef bleaching severity in the same period, acknowledging the circular nature of using observed data to define and then test sensitivity.”

Line 247 and 248, these two sentences seem to directly contradict each other. Perhaps fix by saying "does not agree with SOME previous findings."

We have corrected the contradiction by changing “does not agree with previous findings” to “does not agree with SOME previous findings.”

Line 224

“However, it does not agree with some previous findings^{29,31}.”

Line 273, delete repetitive sentence “Higher bleaching severity...”

We have deleted the repetitive sentence “Higher bleaching severity...”.

Line 275, replace “These” with “Thus, our results...” and line 277, connect the two sentences with “...heat and light attenuation, and agree...”

We replaced “These” with “Thus, our results...” and in line 277 connected the two sentences with “...heat and light attenuation, and agree...”.

Lines 246 to 248:

“Thus, our results contrast with the “depth refugia” hypothesis^{32,33}, which postulates that deep reef areas will be less affected by coral bleaching due to heat and light attenuation, and agrees with other studies that have found no clear depth refuge for coral species^{34,77–80}.”

Line 301 delete “emphasizes the importance of understanding the” and use more direct language like “In conclusion, our study revealed key drivers...”

We have replaced “emphasizes the importance of understanding the” with more direct language: “In conclusion, our study revealed key drivers...”.

Lines 272 and 273:

“In conclusion, our study revealed key drivers of coral bleaching by integrating various reef sensitivity and heat stress metrics.”

Line 305 to 307, this is not a complete sentence. Rephrase. I think what is meant is that the model has both theoretical and practical application for reefs where diversity, depth, historical heat stress, and real-time SST data are available.

We have rephrased the incomplete sentence for clarity, now stating:
Lines 276 to 278:

“The model offers both, theoretical and practical applications, enabling real-time predictions for reefs where data on coral diversity, depth, historical heat stress, and SST data are available.”

Figure 1 – Recommend inserting scale bar and north-pointing arrow.

Figure 1:

As requested, we have added a scale bar and a north-pointing arrow to the figure.

Figure 2 – Caption, line 144 in a) delete “the” and “was”; b) what are the “raw data” shown – are they bleaching severity? If so, what are units, percent? Add units to x-axis “Distance (km)” on the figure and in caption, add distance from what. Is it distance along the MAR going north to south? South to north?

We clarified the description in the caption and added units to the x-axis. We have also specified that the distance is along the MAR and clarified the direction:

Fig. 2. Main drivers of coral bleaching and the identified associations of variation. (a) Relative contribution of the key drivers identified using Gradient Boosted Models (GBMs). (b) Spline correlogram displaying the spatial autocorrelation of the bleaching severity index values (data) and the model residuals. The x-axis represents the distance between sampling points in kilometers, regardless of direction.

Lastly, make sure to update the supplementary material document to reflect suggestions made by the three reviewers and me, especially with regards to variable definitions and explanations. Also, clean up this document to have better resolution figures, consistent numbers of significant digits in tables, and add a cover page with title, authors, etc.

We have updated the supplementary materials to reflect the suggestions from the reviewers, improving figure resolution, ensuring consistent significant digits in the tables, and adding a cover page with the title and authors.